 

# Broadly neutralizing human antibodies against dengue virus identified by single B cell transcriptomics

Natasha D Durham[1†‡], Aditi Agrawal[1†], Eric Waltari[1], Derek Croote[2], Fabio Zanini[2§], Mallorie Fouch[3], Edgar Davidson[3], Olivia Smith[1], Esteban Carabajal[1], John E Pak[1], Benjamin J Doranz[3], Makeda Robinson[4,5], Ana M Sanz[6], Ludwig L Albornoz[7], Fernando Rosso[6,8], Shirit Einav[4,5], Stephen R Quake[1,2], Krista M McCutcheon[1], Leslie Goo[1,9]*

[1]Chan Zuckerberg Biohub, San Francisco, United States; [2]Department of Bioengineering, Stanford University, Stanford, United States; [3]Integral Molecular, Inc, Philadelphia, United States; [4]Division of Infectious Diseases and Geographic Medicine, Department of Medicine, Stanford University School of Medicine, Stanford, United States; [5]Department of Microbiology and Immunology, Stanford University School of Medicine, Stanford, United States; [6]Clinical Research Center, Fundación Valle del Lili, Cali, Colombia; [7]Pathology and Laboratory Department, Fundación Valle del Lili, Cali, Colombia; [8]Department of Internal Medicine, Division of Infectious Diseases, Fundación Valle del Lili, Cali, Colombia; [9]Vaccine and Infectious Disease Division, Fred Hutchinson Cancer Research Center, Seattle, United States

*For correspondence:
lgoo@fredhutch.org

[†]These authors contributed equally to this work

Present address: [‡]Department of Microbiology and Physiological Systems, University of Massachusetts Medical School, Worcester, United States; [§]Lowy Cancer Research Center, University of New South Wales, Kensington, Australia

**Abstract** Eliciting broadly neutralizing antibodies (bNAbs) against the four dengue virus serotypes (DENV1-4) that are spreading into new territories is an important goal of vaccine design. To define bNAb targets, we characterized 28 antibodies belonging to expanded and hypermutated clonal families identified by transcriptomic analysis of single plasmablasts from DENV-infected individuals. Among these, we identified J9 and J8, two somatically related bNAbs that potently neutralized DENV1-4. Mutagenesis studies showed that the major recognition determinants of these bNAbs are in E protein domain I, distinct from the only known class of human bNAbs against DENV with a well-defined epitope. B cell repertoire analysis from acute-phase peripheral blood suggested that J9 and J8 followed divergent somatic hypermutation pathways, and that a limited number of mutations was sufficient for neutralizing activity. Our study suggests multiple B cell evolutionary pathways leading to DENV bNAbs targeting a new epitope that can be exploited for vaccine design.

## Introduction

Dengue virus (DENV) is an enveloped, positive-stranded RNA virus belonging to the *Flavivirus* genus, which includes clinically significant human pathogens such as Yellow Fever virus, Japanese encephalitis virus, West Nile virus (WNV), and Zika virus (ZIKV). DENV is transmitted to humans via *Aedes* mosquitoes, whose global distribution places half of the world's population at risk for infection (*Kraemer et al., 2019*; *Messina et al., 2019*). Each year, the four phylogenetically and antigenically distinct DENV serotypes (DENV1-4) cause approximately 400 million infections (*Bhatt et al., 2013*). Additionally, increased global trade, connectivity, and climate change have fueled the expansion of DENV1-4 into new territories (*Kraemer et al., 2019*; *Messina et al., 2014*).

Approximately 20% of DENV-infected individuals develop a mild febrile illness, of which 5% to 20% progress to potentially fatal severe disease, characterized by bleeding, plasma leakage, shock, and organ failure (*Guzman and Harris, 2015*; *Khursheed et al., 2013*; *Thein et al., 2011*). Epidemiological studies have shown that pre-existing antibodies from a primary DENV infection are a risk factor for severe disease following subsequent infection with a heterologous DENV serotype (*Katzelnick et al., 2017a*; *Salje et al., 2018*; *Sangkawibha et al., 1984*). This is partly attributed to the prevalence of cross-reactive antibodies from the initial infection that can bind, but not neutralize the secondary heterologous virus. Instead, these non-neutralizing antibodies have the potential to facilitate viral uptake into Fc gamma receptor (FcγR)-expressing target cells in a process known as antibody-dependent enhancement (ADE) (*Guzman and Harris, 2015*; *Halstead, 2014*). Recent studies demonstrated that the risk of severe disease following secondary infection is greatest when pre-existing titers of cross-reactive antibodies fall within a narrow, intermediate range (*Katzelnick et al., 2017a*; *Salje et al., 2018*). To limit the potential for ADE, an effective vaccine must therefore elicit durable and potent neutralizing antibodies of high titer against DENV1-4 simultaneously. However, the viral and host determinants leading to such bNAbs against flaviviruses are poorly understood.

All of the leading DENV vaccine candidates in clinical development are based on a tetravalent strategy (*Scherwitzl et al., 2017*), which assumes that the use of representative viral strains from each serotype will elicit a balanced and potent polyclonal antibody response to minimize the risk of ADE. However, the suboptimal efficacy and safety profile of a recently licensed DENV vaccine has been partly attributed to an imbalanced neutralizing antibody response to the four serotypes (*Hadinegoro et al., 2015*). A new tetravalent vaccine candidate in advanced clinical development also displayed serotype-dependent efficacy (*Biswal et al., 2019*). Additionally, there may be important antigenic differences between circulating and lab-adapted strains (*Lim et al., 2019*; *Raut et al., 2019*), as well as among strains even within a given serotype (*Bell et al., 2019*; *Katzelnick et al., 2015*). Antigenic mismatch between vaccine and circulating strains impacted vaccine efficacy (*Juraska et al., 2018*), highlighting the importance of rational selection of vaccine components. An alternative to the tetravalent strategy, largely exemplified by vaccine development efforts for HIV (*Kwong and Mascola, 2018*) and respiratory syncytial virus (RSV) (*Crank et al., 2019*), relies on identifying antibodies with desirable properties and precisely defining their epitopes to guide epitope-based vaccine design (*Graham et al., 2019*). For antigenically diverse viruses such as DENV, a conserved epitope-based vaccine strategy to elicit a broad and potent monoclonal neutralizing antibody response could mitigate the challenge of selecting representative vaccine strains.

The main target of flavivirus neutralizing antibodies is the envelope (E) glycoprotein, which consists of three structural domains (DI, DII, DIII), and is anchored to the viral membrane via a helical stem and transmembrane domain. The E proteins direct many steps of the flavivirus life cycle, including entry, fusion, and assembly of new virus particles (*Pierson and Diamond, 2012*). Flaviviruses bud into the endoplasmic reticulum lumen as immature particles with a spiky surface on which E proteins associate with a chaperone protein, prM (*Prasad et al., 2017*; *Zhang et al., 2003*; *Zhang et al., 2007*). Within the low pH environment of the trans-golgi network, E proteins undergo conformational changes that allow furin-mediated cleavage of prM (*Yu et al., 2008*), resulting in the release of mature infectious virions with a smooth surface densely coated with E homodimers (*Kostyuchenko et al., 2016*; *Kuhn et al., 2002*; *Mukhopadhyay et al., 2003*; *Sirohi et al., 2016*; *Zhang et al., 2013*). The dense arrangement of E proteins on the virion surface is important for antigenicity, as many potently neutralizing human antibodies against flaviviruses target quaternary epitopes spanning multiple E proteins (*de Alwis et al., 2012*; *Hasan et al., 2017*; *Kaufmann et al., 2010*; *Rouvinski et al., 2015*; *Teoh et al., 2012*).

Recent advances in monoclonal antibody isolation and characterization (*Boonyaratanakornkit and Taylor, 2019*; *Corti and Lanzavecchia, 2014*) have accelerated the identification of bNAbs, including those against flaviviruses. Examples include antibodies d488 (*Li et al., 2019*) and m366 (*Hu et al., 2019*), which were cloned from B cells of rhesus macaques receiving an experimental DENV vaccine and from healthy flavivirus-naive humans, respectively, and antibody DM25-3, which was isolated from a mouse immunized with a mature form of DENV2 virus-like particles (*Shen et al., 2018*). Although these antibodies were cross-reactive against DENV1-4, they demonstrated only moderate potency. E protein residues involved in d488 binding lie at the interface of the M protein and the E protein ectodomain (*Li et al., 2019*), while those for m366.6 binding appear to be located at the dimerization interface between DII and DIII (*Hu et al., 2019*). Residue

W101 within the DII fusion loop was important for recognition by antibody DM25-3 (*Shen et al., 2018*). Attempts to engineer mouse antibodies with increased breadth and potency against DENV1-4 have also been described (*Deng et al., 2011*; *Shi et al., 2016*; *Tharakaraman et al., 2013*).

Only a few naturally occurring human bNAbs against flaviviruses have been characterized. Many of these antibodies target epitopes consisting of the highly conserved DII fusion loop as well as the adjacent bc loop in DII in some cases (*Smith et al., 2013*; *Tsai et al., 2013*; *Xu et al., 2017*). Although antibodies recognizing the fusion loop can demonstrate broad reactivity to all DENV sero-types and related flaviviruses, their neutralizing potency is often limited due to this epitope being largely inaccessible, especially on mature virions (*Cherrier et al., 2009*; *Nelson et al., 2008*; *Shen et al., 2018*; *Stiasny et al., 2006*). To date, the most well-characterized class of human anti-bodies with broad and potent neutralizing activity against flaviviruses targets a conserved, quater-nary epitope spanning both E monomeric subunits within the dimer. A subset of these E dimer epitope (EDE)-specific bNAbs potently neutralize not only DENV1-4, but also ZIKV, owing to the high conservation of the EDE, which overlaps the prM binding site on E (*Barba-Spaeth et al., 2016*; *Dejnirattisai et al., 2015*; *Rouvinski et al., 2015*). The exciting discovery of the EDE class of bNAbs highlights the potential for an epitope-focused flavivirus vaccine strategy.

As multiple specificities are likely required to provide maximum coverage of diverse circulating viral variants (*Bell et al., 2019*; *Doria-Rose et al., 2012*; *Goo et al., 2012*; *Katzelnick et al., 2015*; *Keeffe et al., 2018*; *Kong et al., 2015*), in this study, we aimed to define novel sites on the flavivirus E protein that can be targeted by bNAbs. By characterizing 28 monoclonal antibodies from plasma-blasts of two DENV-infected individuals, we identified J8 and J9, clonally related bNAbs that neutral-ized DENV1-4 in the low picomolar range. The major recognition determinants for J8 and J9 were in E protein DI, distinct from previously characterized bNAbs. Analysis of the corresponding B cell rep-ertoire revealed divergent evolution of J8 and J9, suggesting multiple evolutionary pathways to gen-erate bNAbs within this lineage. Our work identifies both viral and host determinants of the development of DENV bNAbs that can guide immunogen design and evaluation.

## Results

### Identification of cross-reactive neutralizing antibodies from clonally expanded plasmablasts of DENV-infected individuals

We previously profiled the single-cell transcriptomics of peripheral blood mononuclear cells (PBMCs) from six dengue patients and four healthy individuals (*Zanini et al., 2018*). From two patients, one (013) with secondary DENV4 infection and another (020) with primary DENV1 infection (*Table 1*), we identified 15 clonal families comprising a total of 38 unique paired heavy (VH) and light (VL) chain IgG1 plasmablast sequences, some of which were hypermutated (1.67% to 10.77% for VH, 0.67% to 7.22% for VL; *Figure 1—figure supplement 1*). One clonal family included members found in both

**Table 1.** Patient characteristics.

|  | Patient 013 | Patient 020 |
| --- | --- | --- |
| Age (years) | 31 | 24 |
| Sex | F | F |
| DENV exposure | Secondary | Primary |
| DENV serotype | 4 | 1 |
| Admission diagnosis[*] | Dengue + warning signs | Dengue |
| Discharge diagnosis[*] | Severe dengue | Dengue |
| Plasmablast sampling time (days post-fever onset) | 4 | 4 |
| Severe disease onset (days post-fever onset) | 5 | n/a |
| Co-morbidities/co-infections/pregnancy | Postpartum day 4 | n/a |

[*] According to 2009 WHO criteria (**WHO, 2009**).

individuals (antibodies B10, M1, and D8 from clonal family 1, *Figure 1—figure supplement 1*), suggesting convergent evolution, which has been described for the antibody response to distinct viruses, including flaviviruses (*Parameswaran et al., 2013*; *Robbiani et al., 2017*), Ebola virus (*Davis et al., 2019*) and HIV (*Scheid et al., 2011*; *Wu et al., 2011*). To functionally characterize these monoclonal antibodies, we successfully cloned 36 paired VH and VL sequences into expression vectors, and transfected mammalian cells for small scale (96-well) recombinant IgG1 production. We detected secreted IgG in the transfection supernatants for 28 of 36 antibodies, which were tested for binding to DENV2 recombinant soluble E protein and reporter virus particles, as well as for neutralizing activity against a panel of reporter flaviviruses, including DENV1-4, ZIKV, and WNV (*Figure 1—figure supplement 1*). Seventeen of 28 antibodies bound to either DENV2 soluble E (n = 1), or reporter virus (n = 7), or both (n = 9). None of the antibodies neutralized ZIKV, but all 28 neutralized at least one DENV serotype. Twenty one antibodies neutralized two or more DENV serotypes, and one antibody neutralized WNV in addition to DENV.

## Binding profile of antibodies

For further characterization, we initially selected six antibodies for larger scale production and IgG purification based on their ability to neutralize at least four of the six viruses tested (*Figure 1—figure supplement 1*). These included two antibody clonal variants found in both patients (B10 from patient 020 and M1 from patient 013), two (C4, J9) and one (L8) patient 013 and patient 020 antibodies, respectively, that neutralized DENV1-4, and the only antibody that neutralized WNV (I7 from patient 020). We first confirmed binding activity at a single antibody concentration (5 µg/ml) by ELISA. Consistent with our pilot screen using crude IgG-containing supernatant (*Figure 1—figure supplement 1*), L8, M1, B10, and I7 bound to both DENV2 soluble E and reporter virus particles, while J9 and C4 bound to the latter only (*Figure 1A–B*), suggesting that these antibodies target epitopes preferentially displayed on the intact virion. As incubation at higher temperatures has been shown to improve exposure of some epitopes (*Dowd et al., 2011*; *Lok et al., 2008*; *Sukupolvi-Petty et al., 2013*), we performed the ELISA at both ambient temperature and 37°C. For most antibodies, incubation at 37°C resulted in a modest but consistent increase in binding to virus particles (*Figure 1B*). To evaluate the relative binding of the antibodies, we also performed a dose-response indirect ELISA at ambient temperature (*Figure 1C*). Binding curves revealed robust binding of antibodies L8, B10, M1, and I7 to DENV2 soluble E ($EC_{50}$ range of 0.4 to 23 ng/ml) while J9, C4 and the EDE antibodies C10 and B7 displayed little to no binding to soluble E even at high antibody concentrations (up to 200 µg/mL). All antibodies bound to DENV2 particles to varying extents, with relatively high $EC_{50}$ values for J9 (200 ng/ml) and C4 (1200 ng/ml), suggesting limited affinity maturation (*Figure 1D*).

## Neutralization potency of antibodies

We next performed dose-response neutralization assays to obtain $IC_{50}$ values (antibody concentration at which 50% of virus infectivity was inhibited). Antibodies M1, B10, and L8 displayed modest (average $IC_{50}$ range against DENV1-4 of 721 to 1670 ng/ml) and incomplete neutralization of DENV1-4, with ~10% to~50% infectivity persisting at the highest antibody concentration tested (10 µg/ml) (*Figure 2A–B*). Incomplete neutralization is commonly observed for cross-reactive DII fusion loop-specific antibodies, and likely represents structurally heterogeneous virions on which the epitope is not displayed frequently enough for antibodies to bind at a stoichiometry sufficient for neutralization (*Nelson et al., 2008*; *Pierson et al., 2007*). Antibody I7 displayed an unusual neutralization profile as it did not neutralize DENV4, and its potency against DENV1-3 was lower than that against the more antigenically distant WNV. Although antibody C4 completely neutralized DENV1-4, it did so with modest potency, especially against DENV4 ($IC_{50}$ >1000 ng/ml). The most potent antibody we identified was J9, which despite relatively weak binding (*Figure 1*), completely neutralized DENV1-4 with average $IC_{50}$ values of 6 ng/ml, 30 ng/ml, 15 ng/ml, and 39 ng/ml, respectively. A previously characterized subgroup of the EDE class of bNAbs, which includes EDE1 C10 can neutralize not only DENV1-4, but also ZIKV (*Barba-Spaeth et al., 2016*). J9 showed a high specificity for DENV, with no activity against ZIKV, and up to ~60 fold greater potency against some DENV serotypes compared to EDE1 C10 (*Figure 2B*). Depending on the serotype, the average neutralization potency of J9 against DENV1-4 was also up to 14-fold higher than that of bNAb EDE2 B7,

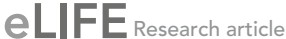

**Figure 1.** Binding profile of antibodies. A single dilution of the antibodies indicated on the x-axis was tested for binding to DENV2 (**A**) soluble E protein and (**B**) reporter virus particles at room temperature (RT) and 37°C by ELISA. The y-axis shows absorbance values at 450 nm (A450). Error bars indicate the range of values obtained in duplicate wells. Data are representative of 3 independent experiments. The dotted horizontal line in (**B**) indicates the average A450 values obtained for negative control WNV-specific antibody CR4354 at 37°C. Representative dose-response binding curves of the indicated antibodies to DENV2 (**C**) soluble E protein and (**D**) reporter virus at room temperature. The y-axis shows binding signal intensity in arbitrary units (AU). Data points and error bars indicate the mean signal intensity and standard deviation (SD) of triplicate spots within one well of the microarray, respectively. Binding curves are representative of two independent experiments.

The online version of this article includes the following figure supplement(s) for figure 1:

**Figure supplement 1.** Characteristics of plasmablast-derived antibodies from DENV-infected patients.

which belongs to another EDE subgroup with poor neutralizing activity against ZIKV (*Barba-Spaeth et al., 2016*).

The broad and potent neutralizing activity of J9 prompted us to re-evaluate our pilot results obtained for J8, a somatic variant with no binding or neutralizing activity in our screen with crude IgG-containing supernatant (*Figure 1—figure supplement 1*). When we repeated the cloning, expression, and purification of J8 IgG, we observed similar binding (*Figure 1*) and neutralization (*Figure 2*) profiles to J9. When tested as Fab fragments, J9 and J8 were still able to potently neutralize DENV, unlike C4 and EDE1 C10, which failed to achieve 50% neutralization at the highest Fab concentration tested (*Figure 2—figure supplement 1*). In addition to potent neutralization of DENV1-4 reporter viruses, which are based on lab-adapted strains isolated between 1964–1982, J9

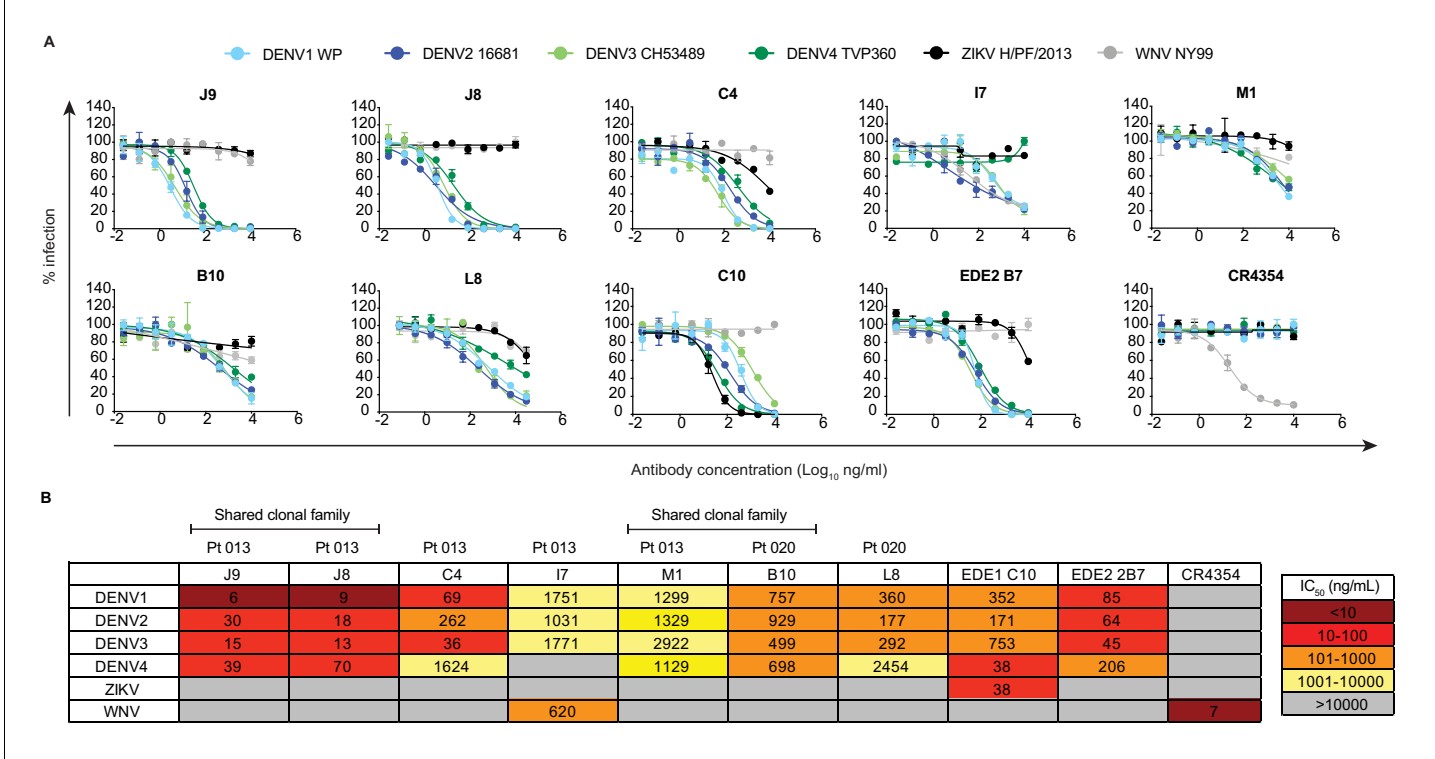

**Figure 2.** Neutralization profile of antibodies. (**A**) Representative antibody dose-response neutralization curves against DENV1-4, ZIKV, and WNV reporter viruses. Infectivity levels were normalized to those observed in the absence of antibody. Data points and error bars indicate the mean and range of duplicate wells, respectively. Results are representative of at least three independent experiments. (**B**) Antibody concentrations resulting in 50% inhibition of infectivity (IC$_{50}$) from dose-response neutralization experiments described in (**A**). Values represent the mean of at least three independent experiments, each performed in duplicate and summarized in *Figure 2—source data 1*. The heatmap indicates neutralization potency, as defined in the key. Gray boxes indicate that 50% neutralization was not achieved at the highest antibody concentration tested (10 µg/ml). The patient (Pt) from which antibodies were isolated are indicated above each antibody name. Antibodies from shared clonal families are indicated above patient ID.

The online version of this article includes the following source data and figure supplement(s) for figure 2:

**Source data 1.** Antibody IC$_{50}$ values against a panel of reporter flaviviruses from replicate experiments.
**Figure supplement 1.** Neutralization potency of IgG and Fab fragments.
**Figure supplement 2.** Neutralization potency of antibodies against fully infectious DENV1-4 strains.
**Figure supplement 3.** Antibody neutralization of standard and mature reporter virus preparations.
**Figure supplement 4.** Mechanism of neutralization.

and J8 also neutralized DENV1-4 strains isolated more recently (between 2004–2007) with IC$_{50}$ values < 50 ng/ml (*Figure 2—figure supplement 2*). Additionally, in contrast to C4 and the cross-reactive DII fusion loop-specific mouse antibody E60 (*Goo et al., 2017*; *Nelson et al., 2008*; *Oliphant et al., 2006*), but similar to EDE bNAbs (*Dejnirattisai et al., 2015*), J9 and J8 potently neutralized DENV regardless of virion maturation state (*Figure 2—figure supplement 3*), which can indirectly modulate epitope exposure (*Cherrier et al., 2009*; *Goo et al., 2019*; *Nelson et al., 2008*) and has been shown to be distinct among circulating versus lab-adapted strains (*Raut et al., 2019*). We also tested the ability of antibodies to mediate neutralization after virus attachment to cells, which is characteristic of many potently neutralizing antibodies against flaviviruses (*Goo et al., 2019*; *Nybakken et al., 2005*; *Sukupolvi-Petty et al., 2010*; *Vogt et al., 2009*; *Xu et al., 2017*). When added after virus attachment to Raji-DCSIGNR cells, C4 failed to inhibit 40–50% of infection at the highest antibody concentration tested (300 µg/ml) (*Figure 2—figure supplement 4*). In contrast, J9, J8, and EDE1 C10 potently inhibited DENV2 infection both pre- and post-virus attachment to cells.

## ADE potential of antibodies

*In vitro*, antibodies can mediate ADE of infection in cells expressing FcγR at sub-neutralizing concentrations (*Pierson et al., 2007*). Recent studies in humans have also demonstrated that the risk of severe dengue disease following secondary infection is greatest within a range of intermediate titers of pre-existing DENV-specific antibodies, while higher titers are protective against symptomatic infection (*Katzelnick et al., 2017a*; *Salje et al., 2018*). Thus, eliciting potently neutralizing antibodies is desirable to limit the concentration range within which ADE can occur. We measured the ADE potential of a subset of antibodies identified above in K562 cells, which have been used extensively to study ADE of flaviviruses as they express FcγR and are poorly permissive for infection in the absence of antibody (*Littaua et al., 1990*). As expected, pre-incubation of DENV with all DENV-specific antibodies resulted in a dose-dependent enhancement of infection to varying extents (*Figure 3A* and *Figure 3—figure supplement 1*). We measured the antibody concentration at which the highest level of ADE was observed (peak enhancement titer). Consistent with their high neutralization potencies, the average peak enhancement titer of J9 and J8 for DENV2 (3 ng/ml) was approximately 27-fold and 480-fold lower than that of antibodies EDE1 C10 (80 ng/ml) and C4 (1467 ng/ml), respectively (*Figure 3A and D*). J9 and J8 also displayed the lowest peak enhancement titers

**Figure 3.** Antibody-dependent enhancement (ADE) of DENV2, ZIKV and WNV infection. Serial dilutions of the antibodies indicated above each graph were pre-incubated with (A) DENV2, (B) ZIKV or (C) WNV reporter virus for 1 hr at room temperature prior to infection of K562 cells, which express FcγR and are poorly permissive for direct infection in the absence of antibodies. The y-axis shows the percentage of infected GFP-positive cells quantified by flow cytometry. Data points and error bars indicate the mean and range of infection in duplicate wells, respectively. Bar graphs represent average antibody concentrations at peak enhancement of (D) DENV2, (E) ZIKV or (F) WNV infection obtained from 2 to 3 independent experiments, each represented by a data point. Where indicated, error bars represent the SD in panels **D-F**.

The online version of this article includes the following figure supplement(s) for figure 3:

**Figure supplement 1.** ADE of DENV1, DENV3, and DENV4 infection.

for DENV1, DENV3, and DENV4 (*Figure 3—figure supplement 1*). For J9, and J8, DENV neutralization occurred beyond the peak enhancement titer, with no infectivity observed at high antibody concentrations. In contrast, for C4 and L8, which neutralized DENV relatively weakly (*Figure 1*), ADE of DENV was still detected at the highest antibody concentration (10 µg/ml) tested (*Figure 3A* and *Figure 3—figure supplement 1A–C*). At high concentrations, L8 also enhanced ZIKV (*Figure 3B*) and WNV infection (*Figure 3C*), suggesting binding, but not neutralizing (*Figure 2*) activity against these viruses. Consistent with their ability to recognize ZIKV, EDE bNAbs enhanced ZIKV infection at sub-neutralizing concentrations (*Figure 3B*). J9 and J8 did not facilitate ADE of ZIKV nor WNV infection, suggesting lack of binding to these flaviviruses. Given their high neutralization potencies against DENV1-4, J9 and J8 represent desirable antibodies to elicit as their ADE potential is restricted to a narrow range of low antibody concentrations, beyond which neutralization is observed.

## Epitope specificity of antibodies

To identify amino acid residues required for antibody recognition, we screened a shotgun alanine-scanning mutagenesis library of DENV2 E protein variants for antibody binding by flow cytometry (*Davidson and Doranz, 2014*). Antibody binding profiles to this entire library are summarized in *Figure 4—source data 1*. Antibodies M1 and L8 demonstrated loss of binding to variants encoding W101A and F108A mutations within the DII fusion loop (*Figure 4A*). Similar to a subset of previously described fusion loop-specific antibodies (*Cherrier et al., 2009*; *Smith et al., 2013*; *Tsai et al., 2013*), M1 recognition also depended on residue G106 within the fusion loop and residue 75 on the adjacent bc loop (*Figure 4A*). Specificity for the fusion loop epitope likely contributes to the inability

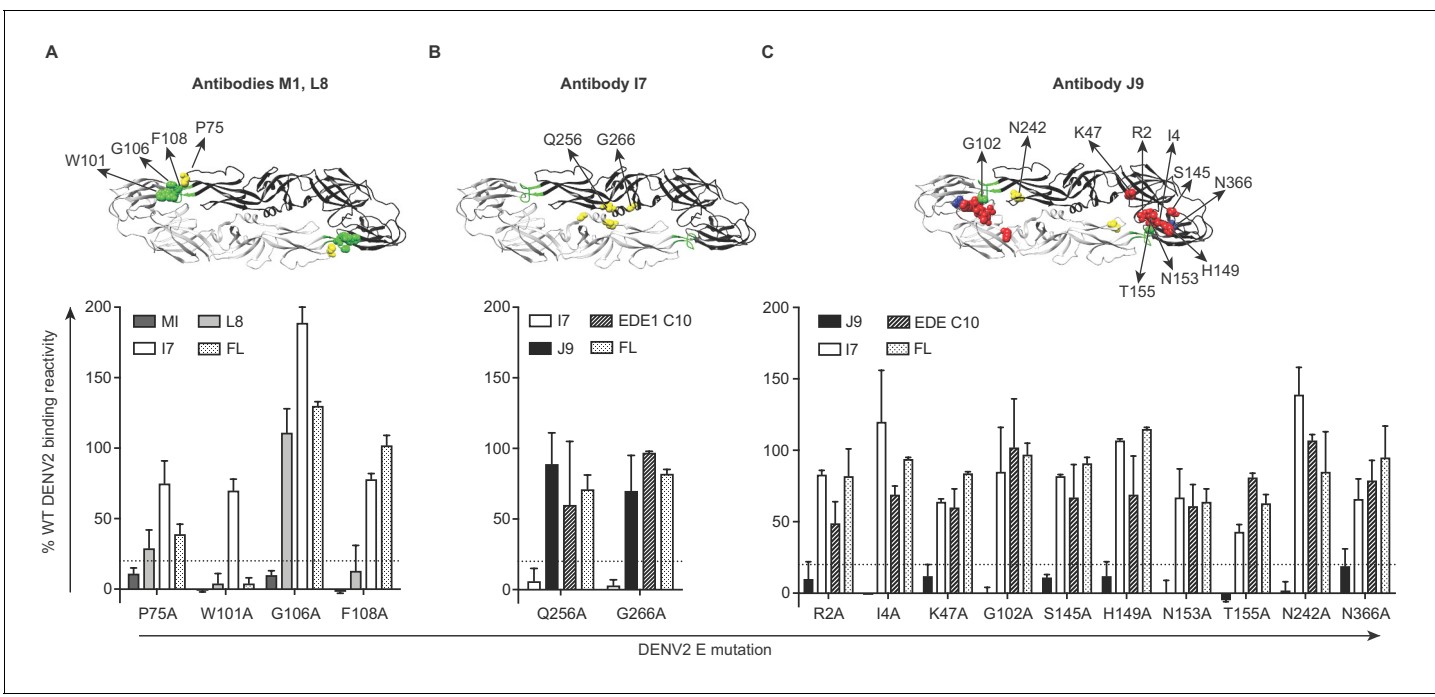

**Figure 4.** Critical E protein residues for antibody binding. Individual alanine mutations of a subset of DENV2 E residues decreased binding by antibodies (A) M1 and L8, (B) I7 or (C) J9 as shown, but did not affect binding by other antibodies, including EDE1 C10 and a previously screened control antibody (FL) targeting the fusion loop (unpublished). Above each graph, residues involved in binding of the indicated antibodies are highlighted on the ribbon structure of one of the monomers (black) within the DENV2 E dimer (PDB: 1OAN). Residues in DI, DII, DIII, and DII fusion loop are indicated in red, yellow, blue, and green, respectively. For each graph, the x-axis indicates the DENV2 E protein mutation and the y-axis displays antibody binding reactivity to the mutant as a percentage of WT DENV2 reactivity. Binding reactivity to the complete mutagenesis library can be found in *Figure 4—source data 1*. Error bars represent the mean and range of at least two independent experiments. The dotted horizontal line indicates 80% reduction in antibody binding reactivity to mutant compared to WT DENV2.

The online version of this article includes the following source data for figure 4:

**Source data 1.** Mean antibody binding reactivity to DENV2 E protein alanine scanning mutagenesis library expressed as a percentage of binding reactivity to wildtype DENV2 from at least two independent experiments.

of these antibodies to neutralize completely, even at high concentrations (*Figure 2A*), as previously described (*Dowd et al., 2011*; *Nelson et al., 2008*). The I7 epitope involved DII residues Q256 and G266 (*Figure 4B*), which are conserved among many flaviviruses (indicated by yellow squares in *Figure 5—figure supplement 1A*) and are important for recognition by the recently described cross-reactive antibody (d448) isolated from vaccinated rhesus macaques (*Li et al., 2019*). Unlike I7, d448 neutralized DENV4, but not WNV (*Li et al., 2019*). Despite testing different temperature and pH conditions, we did not detect C4 binding to WT DENV2 in this flow cytometry-based assay (data not shown) and were thus unable to screen against the mutant library. For J9, most E protein mutations that reduced binding by >80% relative to WT (shown in bold in *Figure 4—source data 1*) were found in DI (R2, I4, K47, S145, H149, N153, T155), but G102 within DII fusion loop and DII N242, as well as DIII N366 were also important (*Figure 4C*). These mutations minimally affected EDE1 C10 binding (*Figure 4C*), suggesting a distinct epitope.

For further epitope mapping of J9, one of the most potent bNAbs we identified, we first attempted to select for neutralization escape viral variants but were unsuccessful after six serial passages in cell culture under antibody selection pressure. We noted that J9 neutralized DENV1-4 but not ZIKV (*Figure 1*). As an alternative epitope mapping approach, we generated a panel of DENV2 reporter virus variants encoding individual mutations at solvent accessible E protein residues on the mature DENV virion that are identical or chemically conserved across representative DENV1-4 strains but differ from ZIKV (*Figure 5—figure supplement 1A*). These residues in the E protein of DENV2 strain 16681 were substituted for analogous residues in ZIKV strain H/PF/2013. We also included a subset of alanine mutants identified by our binding screen to confirm their importance for neutralization. In total, we generated 34 DENV2 reporter virus variants encoding individual mutations throughout the E protein (*Figure 5—figure supplement 1B*); 31 of these variants retained sufficient infectivity (*Figure 5—figure supplement 2A*) for neutralization studies. Most individual mutations displayed minimal (<2 fold) effects on J9 neutralization potency (*Figure 5—figure supplement 3*). Two DI mutations, K47T and V151T resulted in a modest 4-fold increase in IC$_{50}$ (*Figure 5—figure supplement 3*). Consistent with our binding screen, we confirmed that mutation at residue N153 or T155, each of which results in a loss of a potential N-linked glycosylation site (*Figure 5—figure supplement 1A*), abrogated J9 neutralization, while mutation at N242 resulted in a 5-fold reduction in potency (*Figure 5B*). Although our binding screen suggested that individual G102A and S145A mutations contributed to J9 recognition, they had limited effects (~2 fold) on neutralization potency (*Figure 5—figure supplement 3*).

Because we observed only modest effects with single mutations, we next generated DENV2 reporter viruses encoding the K47T and V151T mutations in combination, as well as eight additional pairs of mutations at a subset of the above 34 residues selected based on their proximity to each other on the E dimer structure (*Figure 5—figure supplement 1C*). Seven of 8 DENV2 variants encoding paired mutations retained infectivity (*Figure 5—figure supplement 2B*). Five combinations of paired mutations displayed similarly modest (up to 4-fold) effects on J9 neutralization potency as when these mutations were tested individually (*Figure 5—figure supplement 3A and B*). However, in combination, K47T+V151T reduced J9 potency by almost 100-fold (*Figure 5B*). In combination with the F279S mutation in DI, K47T also resulted in a 16-fold average reduction in J9 potency (*Figure 5B*). *Figure 5A–B* highlight key residues that reduced J9 neutralization potency, either alone or in combination, as identified from our screen against the entire panel of single and double mutants (*Figure 5—figure supplement 3*).

As seen for J9, individual mutations at residues N153 and N155, which together encode a potential N-linked glycosylation site, abrogated the neutralizing activity of the somatically related bNAb, J8 (*Figure 5C*). To varying extents, a similar set of individual and paired mutations that reduced J9 neutralization potency, including V151T, N242A, K47T+V151T, and H149S+V151T, also reduced J8 and C4 neutralization potency (*Figure 5C–D*). However, individual mutations at some DII and DIII residues that did not impact J9 neutralization potency did modestly increase resistance to neutralization by J8 (*Figure 5—figure supplement 3B*) and C4 (*Figure 5—figure supplement 3C*), respectively by 4- to 7-fold. Interestingly, the V151T mutation alone or in combination with either K47T or H149S increased neutralization potency of EDE1 C10 by 25-fold (*Figure 5E*). These mutations had minimal (<2 fold) effects on the neutralizing activity of EDE2 B7 (*Figure 5F*) as well as patient 013 polyclonal sera (*Figure 5—figure supplement 4*), suggesting that they did not globally alter antigenicity. As previously shown (*Dejnirattisai et al., 2015*; *Rouvinski et al., 2015*), mutation at residue

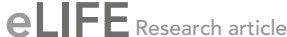

**Figure 5.** Critical E protein residues for antibody neutralization. (A) Ribbon structure of the DENV2 E dimer (PDB: 1OAN) with one monomer in black and the other in gray. The conserved DII fusion loop is shown in green. Colored spheres indicate the location of individual mutations at residues that contribute to J9 recognition based on our screen in *Figure 5—figure supplement 3* and summarized in (B). Bar graphs depict the mean fold change in IC$_{50}$ values against DENV2 reporter virus encoding E protein mutations indicated on the x-axis relative to wildtype DENV2 for antibodies (B) J9, (C) J8, *Figure 5 continued on next page*

*Figure 5 continued*

(**D**) C4, (**E**) EDE1 C10, and (**F**) EDE2 B7. For each antibody, wildtype ZIKV was included as a control. Mean values were obtained from 2 to 7 independent experiments represented by data points. Error bars indicate the standard deviation (n > 2 experiments) or range (n = 2 experiments). Bar colors correspond to those of spheres in (**A**) to indicate location of individual mutations within the E dimer. The locations of paired mutations are shown in *Figure 5—figure supplement 1C*. The dotted horizontal line indicates a 4-fold increase in $IC_{50}$ value relative to wildtype DENV2.

The online version of this article includes the following figure supplement(s) for figure 5:

**Figure supplement 1.** DENV2 E protein mutagenesis.
**Figure supplement 2.** Effect of DENV2 E protein mutations on infectious titer.
**Figure supplement 3.** Effect of E protein mutations on antibody neutralization potency.
**Figure supplement 4.** Effect of E protein mutations on patient 013 serum neutralization potency.

N153 or N155 improved and disrupted EDE1 C10 and EDE2 B7 recognition, respectively (*Figure 5E–F*). Overall, these results suggest that the recognition determinants of J9, J8, and C4, are distinct from those targeted by EDE-specific and other previously characterized bNAbs against DENV1-4 (*Dejnirattisai et al., 2015*; *Hu et al., 2019*; *Li et al., 2019*; *Smith et al., 2013*; *Tsai et al., 2013*; *Xu et al., 2017*).

## Lineage analysis reveals memory origin and divergent evolution of bNAbs

To gain insight into the development of bNAbs J9 and J8, we processed PBMCs of patient 013, from which these bNAbs were identified. We performed next generation sequencing of the B cell receptor repertoire from PBMCs obtained four days post-fever onset. Given its greater junctional diversity compared to light chain, we focused our analysis on the heavy chain repertoire, which is sufficient to identify clonal relationships (*Zhou and Kleinstein, 2019*). We have recently shown that PBMC stimulation in a polyclonal, B cell receptor-independent manner can selectively expand antigen-specific memory B cells (*Waltari et al., 2019*). Accordingly, we obtained 8-fold more unique VH sequences from stimulated PBMCs with a greater representation of IgG over IgM clonal families compared to unstimulated PBMCs (*Table 2*). Compared to previously described healthy B cell

**Table 2.** Summary of Ig sequences, clonal families and their corresponding mean somatic hypermutation (SHM) in unstimulated PBMCs vs. stimulated PBMCs from patient 013.

|  | Unstimulated PBMCs | Stimulated PBMCs |
|---|---|---|
| VH all unique reads | 162928 | 1245789 |
| VH sequences with UMI ≥ 2 | 18588 | 146287 |
| VH clonal families | 11596 | 11407 |
| IgM sequences | 15318 | 4684 |
| SHM, IgM sequences | 0.6% | 2.7% |
| IgG sequences | 2156 | 129636 |
| SHM, IgG sequences | 6.7% | 7.0% |
| IgA sequences | 1114 | 11967 |
| SHM, IgA sequences | 7.0% | 6.2% |
| Clonal family threshold* | 17.8% | 21.4% |
| IgM clonal families | 9794 | 2260 |
| SHM, IgM clonal families | 0.6% | 1.9% |
| IgG clonal families | 1243 | 5657 |
| SHM, IgG clonal families | 6.0% | 5.6% |
| IgA clonal families | 559 | 3490 |
| SHM, IgA clonal families | 6.9% | 6.3% |

VH = variable heavy chain, VL = variable light chain, UMI = unique molecular identifier. * Nucleotide distance used to group related sequences into clonal families.

receptor repertoire data (*Waltari et al., 2019*), VH1-69, VH3-30, VH3-30-3, VH4-34, VH4-39 and VH4-59 were the most dominant across both unstimulated and stimulated PBMC VH families in patient 013 (average >= 5% of repertoire, *Figure 6—figure supplement 1*).

We found 579 and 43,179 VH sequences related to J9/J8 in unstimulated and stimulated PBMCs, respectively (0.4% and 3.5% of total reads, respectively, *Table 3*). For lineage construction (*Figure 6A*), we included sequences that met one of three criteria: 1) highest numbers of unique molecular identifier (UMI) counts (>35 in unstimulated PBMC repertoire, >150 in stimulated PBMC repertoire IgG sequences, and >15 in IgA sequences), 2) <5% somatic hypermutation, or 3) 97% identity to J9 or J8. The J9/J8 lineage derived from recombination of IGHV1-69 with IGHD2-2 and IGHJ5 with no CDRH3 insertions or deletions (*Figure 6B*). The majority of the clonal family members were of the IgG1 subtype, with no IgM identified having a UMI count >2 and only a small percentage of IgA (1.8% of stimulated PBMC relatives; *Figure 6A*, triangles).

We identified clones with a 100% match at the nucleotide level to J9 and J8 in the stimulated PBMC repertoire (UMI counts of 9 and 14, respectively), and related clones identical in both unstimulated and stimulated PBMC repertoires throughout the various branches of the lineage (branch tips labeled H, I, L, N, and O in *Figure 6A*; see *Figure 6—source data 1* for complete nucleotide sequences). Overall, the repertoire showed a rapid expansion of class switched IgG with numerous point mutations from germline VH, strongly suggesting both J8 (27 nucleotide and 18 amino acid mutations) and J9 (30 nucleotide and 25 amino acid mutations) plasmablasts derived from memory B cells from a prior infection, consistent with previous studies (*Priyamvada et al., 2016*; *Xu et al., 2016*). Among this acute-phase repertoire, we did observe less mutated IgG clones A (10 nucleotide and five amino acid mutations), B (13 nucleotide and seven amino acid mutations), and C (12 nucleotide and eight amino acid mutations) early in the lineage (*Figure 6A–B*), which could represent antibodies derived from a *de novo* immune response, or from less mutated memory clones. Finally, the divergent evolution of J9 and J8 suggested multiple somatic hypermutation pathways within this lineage leading to bNAbs.

## VH and VL maturation contributes to broadly neutralizing activity

As described above, J8 and J9 VH derived from V-D-J recombination of IGHV1-69 with IGHD2-2 and IGHJ5. Although we did not analyze the light chain repertoires, both J8 and J9 used the same founder germline IGKV3-11 and IGKJ2 genes with identical CDR lengths and no convergent mutations from germline (*Figure 6C*). To investigate the contribution of somatic hypermutation to broadly neutralizing activity, we generated a panel of recombinant IgG variants, confirmed proper folding (*Figure 7—figure supplement 1*), and tested them for neutralizing activity. As expected, recombinant J8 and J9 IgGs expressing fully germline VH and VL had no neutralizing activity against DENV (*Figure 7A–D*). Similarly, J8 and J9 IgG expressing germline VH paired with the corresponding mature VL, and vice versa, did not neutralize DENV1-4, suggesting that both VH and VL somatic hypermutation contributed to neutralizing activity.

Several amino acid mutations occurred early in the J8/J9 lineage, including CDR-H2 I53F, CDR-H3 T99A/P and D100cH (clones labeled A, B, and C in *Figure 6A*, and alignments in *Figure 6B*), and were retained throughout the continued VH somatic hypermutation. To investigate the VH mutation requirements for broadly neutralizing activity, we generated 'J9_VH5mut' and 'J8_VH5mut' variants containing the above three early VH mutations (I53F, T99P, D100cH) and two additional CDR-H2 mutations (Q61D, K62N) common across different lineage branches (*Figure 6A–B*). We paired J9_VH5mut and J8_VH5mut with the corresponding mature VL to generate recombinant IgGs. These five CDR-H2 and CDR-H3 mutations were sufficient for broadly neutralizing activity of J8 against DENV1-4 (*Figure 7A–D*) and of J9 against DENV1 and DENV2 only (*Figure 7A–B*). Compared to fully mature J9, J9_VH5mut displayed reduced neutralization potency against DENV3 and DENV4 (*Figure 7C–D*), suggesting that additional J9 VH mutations were required for neutralization of these viruses. J9 VL mutations also played a role in neutralization of DENV3 and DENV4 as chimeric IgG expressing J9 mature VH with J8 mature VL displayed less potent neutralization of these viruses (*Figure 7C–D*). Finally, although the FR2 of J9 VH contained a glycine insertion not present in many related clones (*Figure 6B*), this insertion was not necessary for neutralizing activity (*Figure 7A–D*).

**Table 3.** Antibody sequence characteristics from patient 013 unstimulated and stimulated PBMCs related to clonal families of single plasmablasts with reactivity to DENV.

| Clonal family (antibodies) | Germline | Isotype | Reads for all clonotypes | |
|---|---|---|---|---|
| | | | Unstimulated | Stimulated |
| 10 | IGHV3-30-3 | Total | 143 | 216292 |
| (C1, A7, I11, L9, P2, G5) | | IgM | 8 | 427 |
| | | IgG | 112 | 199151 |
| | | IgA | 23 | 16714 |
| 5 | IGHV1-69 | total | 422 | 141308 |
| (C4, J2, N2) | | IgM | 9 | 91 |
| | | IgG | 413 | 140117 |
| | | IgA | 0 | 1100 |
| 13 | IGHV4-34 | total | 32 | 82814 |
| (N8, F4) | | IgM | 0 | 195 |
| | | IgG | 32 | 76605 |
| | | IgA | 0 | 6014 |
| 7 | IGHV1-69 | total | 579 | 43179 |
| (J8, J9) | | IgM | 10 | 98 |
| | | IgG | 568 | 42315 |
| | | IgA | 1 | 764 |
| 9 | IGHV1-69 | total | 673 | 35793 |
| (K11, L3, M4, M11, O4) | | IgM | 12 | 103 |
| | | IgG | 656 | 34444 |
| | | IgA | 5 | 1246 |
| 8 | IGHV1-69 | total | 10 | 7349 |
| (J3, 403_P4) | | IgM | 0 | 16 |
| | | IgG | 10 | 7001 |
| | | IgA | 0 | 332 |
| 3 | IGHV1-18 | total | 10 | 2993 |
| (405_P4, I13) | | IgM | 1 | 6 |
| | | IgG | 9 | 2899 |
| | | IgA | 0 | 88 |
| 1 | IGHV4-39 | total | 4 | 1087 |
| (B10, M1, D8) | | IgM | 1 | 11 |
| | | IgG | 3 | 1049 |
| | | IgA | 0 | 27 |
| 2 | IGHV1-69 | total | 0 | 1087 |
| (H3, M6) | | IgM | 0 | 7 |
| | | IgG | 0 | 1066 |
| | | IgA | 0 | 14 |
| 11 | IGHV3-30-3 | total | 0 | 19 |
| (E9, I8) | | IgM | 0 | 0 |
| | | IgG | 0 | 19 |
| | | IgA | 0 | 0 |

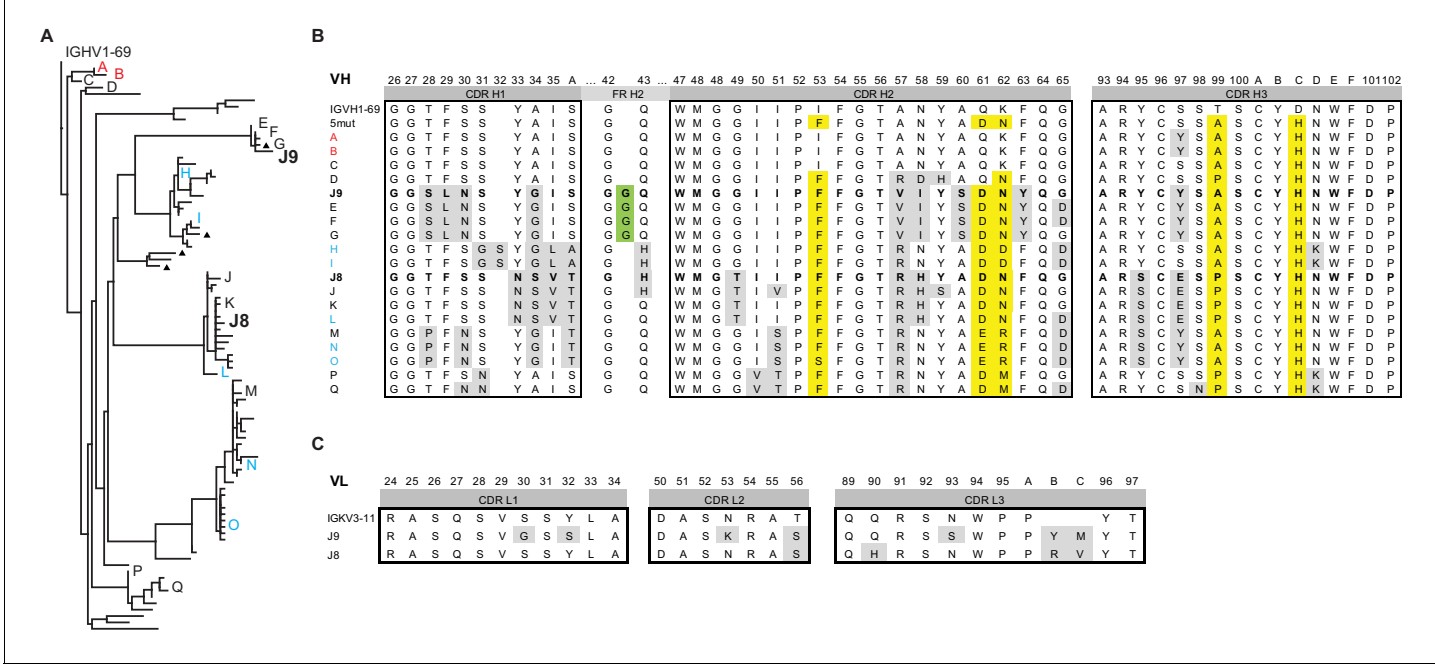

**Figure 6.** Lineage analysis of the J9/J8 clonal family. (A) Maximum likelihood phylogeny of antibodies related to J9 and J8 found in the repertoire of patient 013 was created using the HLP19 model in IgPhyML. VH germline (IGHV1-69*05 + IGHD2-2 + IGHJ5) is shown at the top. Red, black, and blue letters indicate sequences found in unstimulated, stimulated, or both unstimulated and stimulated PBMCs, respectively. Triangles next to tips indicate IgA instead of IgG sequences. (B) Heavy chain CDR alignment of selected antibodies within the J9/J8 lineage found in the repertoire of patient 013. Letters and colors correspond to sequences shown in (A). The germline sequence is shown first, followed by a constructed sequence with five amino acid changes (5mut) highlighted in yellow. A portion FR H2 region is shown to highlight a glycine insertion in clones J9, E, F, and G in green. All other amino acid changes relative to germline are highlighted in gray. Complete VH nucleotide sequences are shown in *Figure 6—source data 1*. (C) Alignment of J8 and J9 light chain CDR to germline sequence (IGKV3-11 + IGKJ2). Amino acid changes relative to germline are highlighted in gray. In both (B) and (C), Kabat numbering is indicated above the alignment.

The online version of this article includes the following source data and figure supplement(s) for figure 6:

**Source data 1.** Characteristics of variable heavy chain nucleotide sequences of selected antibodies in the J9/J8 clonal family.

**Figure supplement 1.** Germline VH gene usage among the unstimulated (circles) and stimulated (triangles) PBMC repertoires of patient 013, with IgM (top) and IgG (bottom) sequences shown.

## Discussion

A safe and effective vaccine to protect against DENV remains elusive, largely due to the challenge of eliciting antibodies that can potently neutralize all four viral serotypes simultaneously to minimize the risk of ADE. Although cross-reactive monoclonal antibodies against flaviviruses have been described, very few display potent neutralizing activity (*Barba-Spaeth et al., 2016*; *Dejnirattisai et al., 2015*; *Xu et al., 2017*). The epitope for one of these bNAbs (SiGN-3C) is not well defined but involves two residues within DII fusion loop and one in DIII (*Xu et al., 2017*). Detailed epitope mapping studies have been performed only for the EDE class of bNAbs, which recognize a quaternary epitope spanning both monomers within the E protein dimer (*Barba-Spaeth et al., 2016*; *Rouvinski et al., 2015*). This epitope involves five main regions on the E protein: *b* strand, fusion loop, and *ij* loop on DII; glycan loop on DI; and the DIII A strand. For antigenically diverse viruses, more than one antibody specificity may be required to provide optimal coverage of diverse circulating variants (*Bell et al., 2019*; *Doria-Rose et al., 2012*; *Goo et al., 2012*; *Katzelnick et al., 2015*; *Keeffe et al., 2018*; *Kong et al., 2015*). In this study, we functionally characterized 28 antibodies identified to be clonally expanded and somatically hypermutated by transcriptomic analyses of single plasmablasts from two individuals acutely infected with DENV (*Table 1*) (*Zanini et al., 2018*). Among these, we identified bNAbs J9 and J8, which potently neutralized all four DENV serotypes and recognize an epitope with major determinants in DI. The location of residues important for J9 and J8 recognition (*Figure 5A*), and the ability of these bNAbs to bind virus

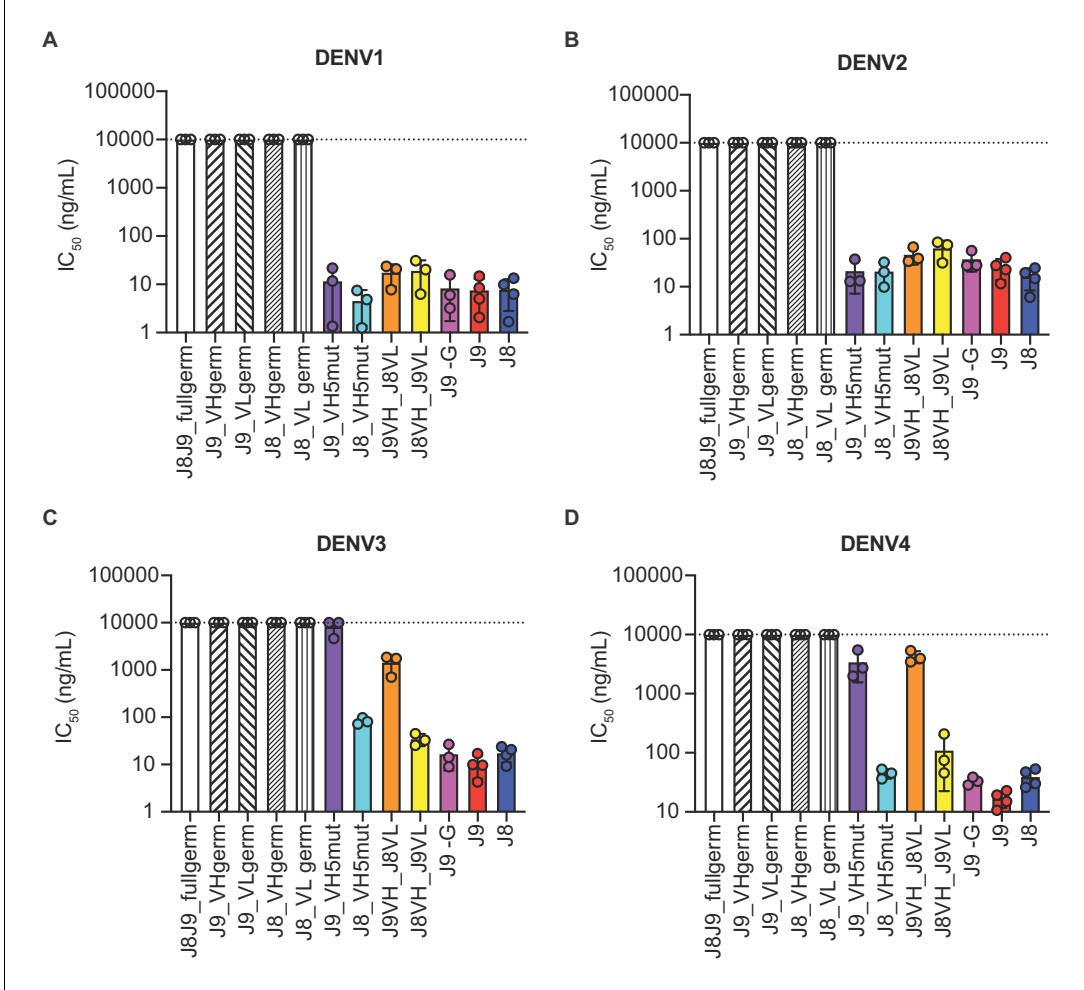

**Figure 7.** Contribution of VH and VL somatic hypermutation to J9 and J8 neutralizing activity. For each antibody indicated on the x-axis, bars represent mean $IC_{50}$ values obtained from 3 to 4 independent experiments indicated by data points against (**A**) DENV1, (**B**) DENV2, (**C**) DENV3, and (**D**) DENV4 reporter viruses. Error bars show the SD. Values at the dotted horizontal line in each graph indicates that 50% neutralization was not achieved at the highest IgG concentration tested (10 µg/ml). $IC_{50}$ values for fully mature J9 and J8 are shown in red and blue bars, respectively. J8J9_full germ: germline J8/J9 VH paired with germline J8/J9 VL; J9_VHgerm: J9 germline VH paired with J9 mature VL; J9VL_germ: J9 mature VH paired with J9 germline VL; J8_VHgerm: J8 germline VH paired with J8 mature VL; J8_VLgerm: J8 mature VH paired with J9 germline VL; J9_VH5mut: J9 VH with five mutations indicated in yellow in **Figure 6B** paired with J9 mature VL; J8_VH5mut: J8 VH with five mutations indicated in yellow in **Figure 6B** paired with J8 mature VL; J9VH_J8VL: J9 mature VH paired with J8 mature VL; J8VH_J9VL: J8 mature VH paired with J9 mature VL; J9 -G: J9 VH with a single glycine deletion in FR2 paired with mature J9 VL.

The online version of this article includes the following figure supplement(s) for figure 7:

**Figure supplement 1.** Analytical size exclusion chromatography of recombinant IgGs.

particles but not soluble E protein (**Figure 1**) suggest a quaternary epitope. Alternatively, the epitope may be localized to the E monomer, but is preferentially displayed on virus particles, as previously described for a DENV1-specific antibody (**Fibriansah et al., 2014**). Nevertheless, our epitope mapping results demonstrated that the recognition determinants for J9/J8 are distinct from EDE bNAbs because E protein mutations that reduced neutralization potency of J9/J8 either increased or did not alter neutralization potency of EDE1 and EDE2 antibodies, respectively (**Figure 5**). Thus, our study defines a new vulnerable site on the DENV E protein that can be exploited for immunogen design to elicit bNAbs.

Future studies relevant to vaccine development include detailed structural characterization to precisely reveal the epitope targeted by J9 and J8, and potentially protein engineering efforts for optimal epitope display (**Graham et al., 2019**). While the detailed characterization of neutralizing

antibodies and their epitopes have not yet informed the design of a successful vaccine, there is promising proof of concept demonstrated for respiratory syncytial virus (*Crank et al., 2019*). In addition to informing vaccine design, J9 and J8 could serve as a foundation for the development of monoclonal antibody therapy, which has recently revolutionized Ebola treatment (*NIAID, 2019*). However, the lack of a robust *in vivo* model of dengue disease continues to hamper our understanding of dengue pathogenesis, and thus the assessment of therapeutic interventions (*Chan et al., 2015*).

A common strategy to isolate and characterize virus-specific antibodies involves sorting hundreds of single B cells from immune donors followed by reverse-transcription (RT)-PCR to isolate paired VH/VL genes for recombinant IgG production and functional characterization (*Dejnirattisai et al., 2015*; *Robbiani et al., 2017*; *Rogers et al., 2017*). Alternatively, single B cells are cultured, and secreted antibodies are directly screened for function (*Walker et al., 2009*). In some cases, memory B cells that specifically bind viral antigens are first enriched by staining with fluorescently labeled antigen (*Cox et al., 2016*; *Robbiani et al., 2017*; *Rogers et al., 2017*; *Scheid et al., 2009*; *Tsioris et al., 2015*; *Woda and Mathew, 2015*; *Wu et al., 2010*). This enrichment step is not feasible when the target antigen is unknown. Additionally, the selection of antigen used as 'bait' may bias the repertoire of recovered antibodies. Although these methods have successfully identified many human bNAbs, including those against flaviviruses (*Dejnirattisai et al., 2015*; *Smith et al., 2013*; *Tsai et al., 2013*; *Xu et al., 2012*; *Xu et al., 2017*), they involve labor intensive steps. Instead of screening a large panel of candidate antibodies, we leveraged transcriptomic analyses of single plasmablasts from acute secondary DENV infection to focus our screen on clonally expanded and somatically hypermutated B cells (*Zanini et al., 2018*), which are likely to encode antigen-specific and affinity matured antibodies. Using this approach, we successfully identified highly potent antibodies capable of neutralizing all four DENV serotypes. It is unclear whether this bioinformatics-based approach to identify DENV bNAbs is fortuitous as acute DENV infection has been shown to induce a rapid and massive expansion of plasmablasts, many of which can neutralize multiple DENV serotypes (*Priyamvada et al., 2016*; *Wrammert et al., 2012*; *Xu et al., 2012*), or whether it is applicable to the rapid identification of highly functional antibodies against other viruses or antigens, as has been suggested in a mouse study (*Reddy et al., 2010*).

J9 and J8 are somatic IgG variants isolated from the same patient (013) who had an acute secondary infection with DENV4 (*Table 1*) (*Zanini et al., 2018*). Next-generation sequencing of the B cell repertoire and phylogenetic analysis of the J9/J8 lineage revealed divergent evolution of these bNAbs (*Figure 6A*), suggesting multiple maturation pathways to generate bNAbs against the J9/J8 epitope, which is encouraging for vaccine design. Antibody lineage divergence and parallel evolution leading to multiple bNAbs within the same individual has also been described in the context of HIV infection (*MacLeod et al., 2016*). Consistent with a previous study of DENV-infected individuals (*Appanna et al., 2016*), many of the DENV-specific antibodies originating from plasmablasts of patient 013, including J9 and J8 also derived from IGVH1-69 (*Table 3*), which is commonly used among bNAbs against other viruses such as influenza and Hepatitis C virus (*Chen et al., 2019*). Many of these bNAbs can achieve neutralization breadth and potency with limited somatic hypermutation (*Lingwood et al., 2012*; *Tzarum et al., 2019*). Despite a moderately high degree of somatic hypermutation for J8 VH (9.9% at the nucleotide level, *Figure 1—figure supplement 1*), five early amino acid mutations in CDR-H2 and CDR-H3 were sufficient for neutralization breadth and potency (*Figure 7*). As the paired VL had a low (1.4%) level of mutation, this observation suggests a relatively limited maturation pathway to a highly functional antibody.

To our knowledge, all human bNAbs against flaviviruses identified in the context of natural infection so far, including J9 and J8, were isolated from plasmablasts of individuals sequentially exposed to at least two DENV serotypes (*Dejnirattisai et al., 2015*; *Xu et al., 2012*; *Zanini et al., 2018*). We also cannot definitively rule out tertiary DENV infection, which is difficult to ascertain in endemic regions, given extensive cross-reactive antibody responses (*Alvarez et al., 2006*; *Bhoomiboonchoo et al., 2015*). Sequential infection with heterologous DENV serotypes or HIV strains has been shown to broaden and strengthen the polyclonal neutralizing antibody response (*Cortez et al., 2012*; *Patel et al., 2017*; *Tsai et al., 2015*; *Tsai et al., 2013*). One proposed model is that low affinity, cross-reactive antibody secreting B-cell clones elicited by primary DENV exposure are reactivated during secondary infection to undergo further affinity maturation resulting in antibodies with more broad and potent neutralizing activity (*Patel et al., 2017*). Although the relatively

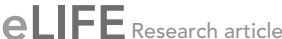

high level of somatic hypermutation already present in J9 and J8 at day four post-fever onset following secondary DENV infection suggests a recall response, repertoire analysis from earlier time points would be required to determine whether the memory B cell clones from which these bNAbs were derived underwent further mutation to achieve neutralization breadth and potency.

It is also unclear whether the specificities and functions of plasmablast-derived antibodies present during acute infection confer long-lived protection from infection and pathogenesis. Despite the presence of plasmablast-derived bNAbs such as J9 and J8, and those belonging to the EDE class during acute secondary infection, the donors from which these antibodies were isolated developed severe dengue disease (*Dejnirattisai et al., 2015*; *Zanini et al., 2018*). J9 and J8 likely appeared too late to mitigate pathogenesis as they were derived from plasmablasts 4 days after onset of fever, and just one day prior to diagnosis of severe disease (*Table 1*). We also note the postpartum status of the patient from which these bNAbs were derived, which could affect overall immune status. The rapid and massive plasmablast activation following acute DENV infection is also coincident with the onset of severe symptoms and has been proposed to contribute to immunopathology (*Wrammert et al., 2012*). Alternatively, despite broad and potent *in vitro* neutralizing activity against a surrogate panel of DENV1-4 strains, these bNAbs may not efficiently neutralize circulating infecting strains. Our data also suggest that J9/J8-like bNAbs make up a minor component of the overall polyclonal antibody response, as they minimally contribute to the neutralizing activity of patient 013 serum (*Figure 5—figure supplement 4*). Additionally, B cell receptor repertoire analysis revealed that though expanded, the J9/J8 clonal family is not the largest in this donor (*Table 3*), at least not in the acute phase sample tested. Although based on the analysis of a single donor, one implication of our finding is that immunogen design to elicit bNAbs may require masking of immunodominant epitopes targeted by weakly neutralizing antibodies to focus the antibody response on the desired epitope (*Weidenbacher and Kim, 2019*). The complex interplay among B cells and antibodies of different specificities and functions, and their impact on immunity and pathogenesis warrant further study, in addition to exploring other potential immunological correlates of protection (*Katzelnick et al., 2017b*), including cell-mediated (*Ng et al., 2019*; *Rothman et al., 2015*) and non-neutralizing antibody responses (*Sharma et al., 2019*).

## Materials and methods

### Patient samples
The study was approved by the Stanford University Administrative Panel on Human Subjects in Medical Research (Protocol #35460) and the Fundación Valle del Lili Ethics committee in biomedical research (Cali, Colombia). All subjects, their parents, or legal guardians provided written informed consent, and subjects between 6 to 17 years of age and older provided assent. We collected blood samples from individuals who presented with symptoms compatible with dengue between 2016 and 2017 to the Fundación Valle del Lili in Cali, Colombia. Cohort details have been previously described (*Zanini et al., 2018*).

### Monoclonal antibodies
Plasmablast-derived variable heavy or light chain sequences (*Zanini et al., 2018*) were synthesized as gene fragments (Genewiz, San Francisco, CA; Integrated DNA Technologies, Coralville, IA) to include at least a 15 basepair overlap with the 5' signal sequence and 3' constant region of our human IgG1, kappa or lambda expression vectors described elsewhere (*Waltari et al., 2019*). For pilot ELISAs and neutralization assays using crude IgG-containing supernatant, paired heavy and light chain plasmids for each antibody were expressed in Expi293F cells (Cat# A14527, ThermoFisher Scientific, Waltham, MA) in a 96-well format. IgG levels were quantified by ELISA as described (*Waltari et al., 2019*). Antibodies with crude IgG expression levels < 0.5 ng/mL were excluded from further characterization. Antibodies selected for in-depth characterization (J9, J8, C4, B10, M1, L8 and I7), as well as control antibodies EDE1 C10 (*Dejnirattisai et al., 2015*), EDE2 B7 (*Dejnirattisai et al., 2015*), CR4354 (*Kaufmann et al., 2010*) were expressed by transient transfection of Expi-CHO-S cells (Cat# A29129; ThermoFisher Scientific). Variable heavy and light chain sequences of the above control antibodies used for gene synthesis and cloning into expression vectors were based on PDB IDs 4UT9, 4UT6, and 3N9G, respectively. Cell culture supernatant was

 

clarified by centrifugation at 3900 xg for 30 min at 4°C, passed through a 0.22 μm filter, and IgG was purified on MabSelect SuRe resin (Cat# 17-5438-01; GE Healthcare, Chicago, IL). Other control antibodies used in this study were obtained commercially: Anti-Dengue Virus Type II Antibody, clone 3H5-1 (Cat# MAB8702; Millipore Sigma, Burlington, MA); Flavivirus group antigen Antibody (D1-4G2-4-15 (4G2)) (Cat# NBP2-52709-0.2mg; Novus Biologicals, Centennial, CO). Mouse monoclonal antibody E60 was provided by Michael Diamond (Washington University, St. Louis, MO).

## Cell lines

Expi293F cells (Cat# A14527; ThermoFisher Scientific) were cultured in Expi293 Expression Medium (Cat# A1435101; ThermoFisher Scientific) according to the manufacturer's instructions. Expi-CHO-S Cells (Cat# A29127; ThermoFisher Scientific) were cultured in ExpiCHO Expression Medium (Cat# A2910001; ThermoFisher Scientific). HEK-293T/17 cells (ATCC CRL-11268) were maintained in DMEM (Cat# 11965118; ThermoFisher Scientific) supplemented with 10% fetal bovine serum (Cat# FB-11; Omega Scientific, Inc) and 100 U/mL penicillin-streptomycin (Cat# 15140–122; ThermoFisher Scientific). Raji cells stably expressing DCSIGNR (Raji-DCSIGNR) (*Davis et al., 2006*) (provided by Ted Pierson, NIH) and K562 cells (ATCC Cat# CCL-243) were maintained in RPMI 1640 supplemented with GlutaMAX (Cat# 72400–047; ThermoFisher Scientific), 10% FBS and 100 U/mL penicillin-streptomycin. All cell lines were maintained at 37°C in 5% $CO_2$ unless otherwise stated. C6/36 cells (ATCC CRL-1660) were maintained in EMEM (ATCC Cat# 30–2003) supplemented with 10% FBS at 30°C in 5% $CO_2$.

All human-derived cell lines (Expi293F, HEK-293T/17, Raji, K562) were authenticated by STR profiling at the Fred Hutchinson Cancer Research Center Specimen Processing/Research Cell Bank core. All cell lines were also tested for mycoplasma contamination using the PCR Mycoplasma Test Kit I/C (Cat # PK-CA91-1024, Promocell, Heidelberg, Germany) and confirmed to be mycoplasma negative.

## Production of reporter virus particles

Reporter virus particles were produced by co-transfection of HEK-293T/17 cells with (i) a plasmid expressing a WNV subgenomic replicon encoding GFP in place of structural genes (*Pierson et al., 2006*), and (ii) a plasmid encoding C-prM-E structural genes from the following viruses: DENV1 Western Pacific (WP) (*Ansarah-Sobrinho et al., 2008*), DENV2 16681 (*Ansarah-Sobrinho et al., 2008*), WNV NY99 (*Pierson et al., 2006*), and ZIKV H/PF/2013 (*Dowd et al., 2016*). Briefly, $8 \times 10^5$ HEK-293T/17 cells pre-plated in a 6-well plate were co-transfected with a mass ratio of 1:3 replicon: C-prM-E plasmids using Lipofectamine 3000 (Cat# L3000-015; ThermoFisher Scientific). Four hours post-transfection, media was replaced with low-glucose DMEM (Cat# 12320–032; ThermoFisher Scientific) containing 10% FBS and 100 U/mL penicillin-streptomycin (i.e. low-glucose DMEM complete) and cells were transferred to 30°C in 5% $CO_2$. Virus-containing supernatant was harvested at 3, 4, and 5 days post-transfection, passed through a 0.22 μm filter, pooled, and stored at −80°C. DENV3 strain CH53489 (Cat# RVP-301; Integral Molecular, Philadelphia, PA) and DENV4 strain TVP360 reporter viruses (Cat# RVP-401; Integral Molecular) were obtained commercially and were produced by co-transfection of the DENV3 or DENV4 CprME plasmid with the DENV2 strain 16681 replicon as previously described (*Mattia et al., 2011*). Reporter virus particles with increased efficiency of prM cleavage were produced as above by co-transfecting plasmids encoding the replicon, structural genes, and human furin (provided by Ted Pierson, NIH) at a 1:3:1 mass ratio. Where indicated, virus was concentrated by ultracentrifugation through 20% sucrose at 164,000 xg for 4 hr at 4°C, resuspended in HNE buffer (5 mM HEPES, 150 mM NaCl, 0.1 mM EDTA, pH 7.4), and stored at −80°C.

Infectious titers of reporter viruses were determined by infection of Raji-DCSIGNR cells. At 48 hr post-infection, cells were fixed in 2% paraformaldehyde (Cat# 15714S; Electron Microscopy Sciences, Hatfield, PA), and GFP positive cells quantified by flow cytometry (Intellicyt iQue Screener PLUS, Sartorius AG, Gottingen, Germany).

## Production, titer and neutralization of fully infectious DENV1-4 isolates

The following DENV strains were used to infect C6/36 cells: DENV1 UIS 998 (isolated in 2007, Cat# NR-49713; BEI), DENV2 US/BID-V594/2006 (isolated in 2006, Cat# NR-43280; BEI), DENV3/US/BID-V1043/2006 (isolated in 2006, Cat# NR-43282; BEI), DENV4 strain UIS497 (isolated in 2004, Cat# NR-49724; BEI). Virus-containing supernatant was collected at days 2–7 post-infection, passed

through a 0.22µm filter, and stored at −80°C. Infectious titer was determined on Raji-DCSIGNR cells. At 48 hr post-infection, intracellular staining was performed using BD Cytofix/Cytoperm Solution Kit (Cat# 554714; BD Biosciences, San Jose, CA) according to the manufacturer's instructions. Mouse antibody 4G2 conjugated to Alexa Fluor 488 (Cat# A20181; Thermo Fisher Scientific) was used for intracellular staining to detect E protein in infected cells by flow cytometry (Intellicyt iQue Screener PLUS, Sartorius AG). Neutralization assays were performed as described below, using intracellular staining with Alexa Fluor 488-conjugated 4G2 to detect infected cells.

## Generation of E protein variants

The DENV2 16681 C-prM-E expression construct (*Ansarah-Sobrinho et al., 2008*) was used as a template for site-directed mutagenesis using the *Pfu* Ultra DNA polymerase system (Cat# 600380; Agilent Technologies, Santa Clara, CA) and primers generated by QuikChange Primer Design (Agilent Technologies). The entire C-prM-E region was sequenced (Quintara, San Francisco, CA) to confirm the presence of the desired mutation(s).

## Shotgun mutagenesis epitope mapping

A DENV2 strain 16681 prM/E expression construct was subjected to high-throughput shotgun mutagenesis to generate a comprehensive mutation library, with each prM/E polyprotein residue mutated to alanine (with alanine residues to serine). In total, 559 DENV2 mutants were generated (99.6% coverage of the prM/E protein), sequence confirmed, and arrayed into 384-well plates (one mutation per well). For antibody library screening, plasmids encoding the DENV protein variants were transfected individually into human HEK-293T cells and allowed to express for 22 hr before fixing cells in 4% paraformaldehyde (Electron Microscopy Sciences), and permeabilizing with 0.1% (w/v) saponin (Sigma-Aldrich, St. Louis, MA) in PBS plus calcium and magnesium (PBS++). Cells were incubated with purified antibodies (0.1–2.0 µg/mL) diluted in 10% normal goat serum (Sigma-Aldrich)/0.1% saponin, pH 9.0. Antibody J9 was screened in unfixed cells that had been co-transfected with the prM/E library and furin expression plasmids, to decrease levels of prM in the cells. Before screening, the optimal concentration was determined for each antibody, using an independent immunofluorescence titration curve against wild-type prM/E to ensure that signals were within the linear range of detection and that signal exceeded background by at least 5-fold. Antibodies were detected using 3.75 µg/mL Alexa Fluor 488-conjugated secondary antibody (Jackson ImmunoResearch, West Grove, PA) in 10% normal goat serum/0.1% saponin. Cells were washed three times with PBS++/0.1% saponin followed by two washes in PBS. Mean cellular fluorescence was detected using a high through-put flow cytometer (Intellicyt iQue Screener Plus, Sartorius AG). Antibody reactivity against each mutant protein clone was calculated relative to reactivity with wild-type prM/E, by subtracting the signal from mock-transfected controls and normalizing to the signal from wild-type protein-transfected controls. The entire library data for each antibody was compared to the equivalent data from control antibodies. Mutations were identified as critical to the antibody epitope if they did not support reactivity of the test antibody (<20% of reactivity to WT prM/E) but supported reactivity of appropriate control antibodies (>70% of reactivity to WT prM/E). This counter-screen strategy facilitates the exclusion of DENV prM/E protein mutants that are mis-folded or have an expression defect (*Davidson and Doranz, 2014*; *Paes et al., 2009*).

## ELISA

High-binding 96-well plates (Cat# CLS3361; Millipore Sigma, Burlington, MA) were either coated directly with 500 ng/well of recombinant DENV2 16681 E protein (Cat# DENV2-ENV-500; The Native Antigen Company, Oxford, UK) or with 300 ng/well of murine antibody 4G2 for capture of concentrated and partially purified reporter DENV2 particles. Recombinant E or capture antibody was added in 100 µl 1X PBS and incubated at 4°C overnight. The following day, 300 µl 1% BSA in PBS blocking buffer (Cat# B0101; Teknova, Hollister, CA) was added for 1 hr either at room temperature (RT) or 37°C. Plates were subsequently washed 6 times using 300 µl PBST (PBS + 0.05% Tween-20) and 100 µl of DENV2 particles diluted 1:10 in blocking buffer was added to wells coated with murine antibody 4G2 and incubated for 1 hr at RT and 37°C. Plates were washed six times and incubated with 5 µg/well of primary antibodies in 100 µl blocking buffer for 1 hr at room temperature or 37°C. Plates were washed six times and incubated with horseradish peroxidase (HRP)-conjugated mouse

anti-human IgG Fc secondary antibody (Cat# 05–4220; ThermoFisher Scientific) diluted 1:1000 in 100 µl blocking buffer for 1 hr at room temperature or 37°C. Plates were washed six times, and 100 µl TMB substrate (Cat# 34028; Thermo Fisher Scientific) was added at room temperature. The reaction was stopped after 6 min by adding 50 µl of 1N HCL. The absorbance at 450 nm was determined using a microplate reader (SpectraMax i3, Molecular Devices, San Jose, CA).

## Protein array printing and ELISA

Spotted protein arrays (5 × 6) were printed onto each well of Greiner high-binding 96-well plates (Cat# 655097, Thermo Fisher Scientific) using a sciFLeXARRAYER S12 (Scienion AG, Berlin). An array with 75x final concentration of sucrose-purified reporter DENV2 particles and a final concentration of 180 µg/mL of recombinant DENV2 16681 E protein was printed alongside 10 µg/mL anti-human IgG Fc (Cat# 09-005-098; Jackson ImmunoResearch), and 1 µg/mL biotinylated kappa secondary (Cat# 2060–08, SouthernBiotech, Birmingham, AL). Probes were diluted 1:1 with D12 buffer (Cat# CBP-5436–25; Scienion AG) and printed at the final concentrations indicated above in triplicate spots from a 384-well source plate (Cat# CPG-5502–1; Scienion AG) chilled to dew point with 3 × 350 pL drops per spot at 60% humidity on each 96-well plate. Plates were cured overnight at 70% humidity before vacuum sealing.

For ELISAs, printed 96-well plates were washed once with binding buffer (0.5% BSA + 0.025% Tween in PBS) then blocked in 100 µl/well blocking buffer (3% BSA in PBS + 0.05% Tween-20 (PBST)). After 1 hr, blocking buffer was removed and 100 µl/well of test antibodies diluted in binding buffer added and incubated overnight at 4°C. We tested twelve 3-fold serial dilutions of J9, J8, C4, EDE1 C10, EDE2 B7 and CR4354 starting at 200 µg/mL; and B10, M1, L8 and I7 starting at 2 µg/mL. The following day, plates were washed three times with PBST, and 100 µl/well of goat anti-human IgG Fc-BIOT (Cat# 2014–08; Southern Biotech) secondary antibody diluted 1:10,000 in binding buffer was added. After 1 hr shaking incubation at room temperature, plates were washed three times, followed by 1 hr shaking incubation at room temperature with 100 µl/well Pierce High Sensitivity Streptavidin-HRP (Cat# 21130, Thermo Fisher Scientific). Plates were again washed three times, developed for 20 min with 50 µl/well SciColor T12 (Cat# CD-5600–100; Scienion AG), then analyzed using sciREADER CL2 (Scienion AG). Dose-response binding curves were analyzed by non-linear regression with a variable slope (GraphPadPrism v8, GraphPad Software Inc, San Diego, CA).

## Neutralization and antibody-dependent enhancement assays

Reporter virus stocks diluted to 5–10% final infectivity (approximately 1000 infectious units) were incubated with 5-fold dilutions of antibody or heat-inactivated serum (56°C for 30 min) for 1 hr at room temperature before addition of $2 \times 10^5$ Raji-DCSIGNR cells (neutralization assays) or K562 cells (ADE assays). The latter express FcγR and are poorly permissive for direct infection in the absence of antibodies, making them a useful model system within which to study ADE *in vitro* (*Littaua et al., 1990*). After 48 hr incubation at 37°C, cells were fixed in 2% paraformaldehyde and GFP positive cells were quantified by flow cytometry (Intellicyt iQue Screener Plus, Sartorius AG). Dose-response neutralization curves were normalized to no antibody controls and analyzed by non-linear regression with a variable slope and the bottom and top constrained to 0% and 100%, respectively (GraphPadPrism v8, GraphPad Software Inc). Fab fragments were generated and purified from IgG using the Pierce Fab Preparation Kit (Cat# PI44985; Thermo Scientific) and used in neutralization assays at 2x molar concentration relative to IgG.

Pre- and post-attachment neutralization assays were carried out as previously described (*Xu et al., 2017*) using Raji-DCSIGNR cells, reporter DENV2, and serial 5-fold dilutions of antibody starting at 150 µg/mL (J9 and J8) or 300 µg/mL (C4 and EDE1 C10). All cells, viruses, antibodies, and media were pre-chilled to 4°C prior to use. For the pre-attachment assay, antibody dilutions were mixed with undiluted DENV2 for 1 hr at 4°C followed by the addition of Raji-DCSIGNR cells and incubation for 1 hr at 4°C. Cells were washed three times with media, resuspended in media and incubated for 48 hr at 37°C. For the post-attachment assay, undiluted DENV2 was incubated with Raji-DCSIGNR cells for 1 hr at 4°C, washed three times with media, resuspended in fresh media and incubated with antibody dilutions. After 1 hr at 4°C, cells were washed three times with media, resuspended in media and incubated for 48 hr at 37°C. After 48 hr at 37°C, cells from both pre- and post-

attachment assays were fixed in 2% paraformaldehyde and GFP positive cells were quantified by flow cytometry, as described above.

## Preparation of PBMCs for B cell receptor repertoire analysis

A 1 ml vial of PBMCs was thawed rapidly in a 37°C water bath, immediately diluted into 9 ml of B cell growth media containing Corning DMEM [+] 4.5 g/L glucose, sodium pyruvate [-] L-glutamine (VWR International, Radnor, PA), 1x Pen/Strep/Glu and 10% ultralow IgG HI-FBS (Thermo Fisher Scientific), and pelleted at 350 xg for 5 min. The cells were resuspended in 1 mL of growth media and filtered through a 5 ml polystyrene tube with a cell strainer cap (Thomas Scientific, Swedesboro, NJ). One half of the PBMCs were transferred to the T25 flask with feeder cells and B cell stimulation media as described previously (*Waltari et al., 2019*) and the other half was spun down in a 1.5 ml Eppendorf tube at 8000 rpm for 5 min, resuspended in 600 µl RLT (Qiagen, Hilden, Germany) + beta-mercaptoethanol, allowed to lyse for 5 min, snap frozen on dry ice and stored at −80°C until RNA purification with the Qiagen AllPrep RNA/DNA kit (Qiagen). Immunoglobulin amplicon preparation, sequencing and analysis were previously described (*Waltari et al., 2019*). Lineage trees were constructed using the IgPhyML package in the Immcantation pipeline (*Hoehn et al., 2016*) to construct a somatic hypermutation-optimized maximum likelihood phylogeny of the heavy chain sequences clonally related to J9 and J8.

## Acknowledgements

We thank the cohort participants and staff; Ted Pierson for providing Raji-DCSIGNR cells and constructs for reporter virus production; Michael Diamond for providing mouse monoclonal antibody E60; Anna Sellas, Gorica Margulis, Esther Ho, and Purnima Ravisankar for lab management and support; Peter Kim and Don Ganem for helpful discussion; Erick Matsen and Duncan Ralph for valuable comments on the manuscript.

Molecular graphics and analyses of the DENV2 E dimer were performed with UCSF Chimera, developed by the Resource for Biocomputing, Visualization, and Informatics at the University of California, San Francisco, with support from NIH P41-GM103311.

Fully infectious DENV1-4 isolates were obtained through BEI Resources, NIAID, NIH, as part of the World Reference Center for Emerging Viruses and Arboviruses program (WRCEVA).

## Additional information

### Competing interests

Eric Waltari, Derek Croote, Fabio Zanini, Shirit Einav, Stephen R Quake, Krista M McCutcheon, Leslie Goo: Inventor of the following patent application, which is co-owned by the Chan Zuckerberg Biohub and Stanford University: PCT patent application entitled ANTIBODIES AGAINST DENGUE VIRUS AND RELATED METHODS, Serial no. PCT/US2019/045427, filed August 7, 2019. Mallorie Fouch, Edgar Davidson: Employee of Integral Molecular. Benjamin J Doranz: Employee and shareholder of Integral Molecular. The other authors declare that no competing interests exist.

### Funding

| Funder | Grant reference number | Author |
|---|---|---|
| Chan Zuckerberg Biohub | | Natasha D Durham<br>Aditi Agrawal<br>Eric Waltari<br>Fabio Zanini<br>Olivia Smith<br>Esteban Carabajal<br>John E Pak<br>Stephen R Quake<br>Krista M McCutcheon<br>Leslie Goo |
| Fred Hutchinson Cancer Research Center | | Leslie Goo |

| National Institutes of Health | HHSN272201400058C | Benjamin J Doranz |
|---|---|---|
| National Science Foundation | Graduate Research Fellowship | Derek Croote |
| Stanford University | Kou-I Yeh Stanford Graduate Fellowship | Derek Croote |
| Dr. Ralph and Marian Falk Medical Research Trust | Catalyst Award | Shirit Einav |
| Stanford Bio-X | Interdisciplinary Initiatives Seed Grants Program | Shirit Einav |
| Stanford University | Stanford Advanced Residency Training at Stanford Fellowship Program | Makeda Robinson |

The funders had no role in study design, data collection and interpretation, or the decision to submit the work for publication.

### Author contributions
Natasha D Durham, Aditi Agrawal, Data curation, Formal analysis, Validation, Investigation, Visualization, Methodology; Eric Waltari, Data curation, Software, Formal analysis, Validation, Investigation, Visualization, Methodology; Derek Croote, Fabio Zanini, Conceptualization, Data curation, Methodology; Mallorie Fouch, Edgar Davidson, Data curation, Formal analysis, Investigation; Olivia Smith, Esteban Carabajal, Investigation; John E Pak, Resources, Validation, Visualization; Benjamin J Doranz, Supervision; Makeda Robinson, Ana M Sanz, Ludwig L Albornoz, Fernando Rosso, Resources; Shirit Einav, Resources, Data curation, Funding acquisition; Stephen R Quake, Conceptualization, Supervision, Funding acquisition; Krista M McCutcheon, Resources, Data curation, Formal analysis, Supervision, Funding acquisition, Validation, Investigation, Visualization, Methodology; Leslie Goo, Conceptualization, Resources, Data curation, Formal analysis, Supervision, Funding acquisition, Validation, Investigation, Visualization, Methodology, Project administration

### Author ORCIDs
Derek Croote (iD) http://orcid.org/0000-0003-4907-1865
Fabio Zanini (iD) https://orcid.org/0000-0001-7097-8539
Shirit Einav (iD) http://orcid.org/0000-0001-6441-4171
Stephen R Quake (iD) http://orcid.org/0000-0002-1613-0809
Leslie Goo (iD) https://orcid.org/0000-0002-3866-6735

### Ethics
Human subjects: The study was approved by the Stanford University Administrative Panel on Human Subjects in Medical Research (Protocol #35460) and the Fundación Valle del Lili Ethics committee in biomedical research (Cali, Colombia). All subjects, their parents, or legal guardians provided written informed consent, and subjects between 6 to 17 years of age and older provided assent.

### Decision letter and Author response
Decision letter https://doi.org/10.7554/eLife.52384.sa1
Author response https://doi.org/10.7554/eLife.52384.sa2

## Additional files

### Supplementary files
• Transparent reporting form

### Data availability
All data generated or analysed during this study are included in the manuscript and supporting files. Source data files have been provided for Figures 2, 4, and 6.

The following previously published dataset was used:

| Author(s) | Year | Dataset title | Dataset URL | Database and Identifier |
|---|---|---|---|---|
| Zanini F, Robinson ML, Croote D, Sahoo MK, Sanz AM, Ortiz-Lasso E, Albornoz LL, Suarez FR, Montoya JG, Goo L, Pinsky BA, Quake SR, Einav S | 2019 | In vivo molecular signatures of severe dengue infection revealed by viscRNA-Seq | https://www.ncbi.nlm.nih.gov/geo/query/acc.cgi?acc=GSE116672 | NCBI Gene Expression Omnibus, GSE116672 |

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
