## [Decision Letter]

Thank you for submitting your work entitled "Broadly neutralizing human antibodies against dengue virus identified by single B cell transcriptomics" for consideration by *eLife*. Your article has been reviewed by three peer reviewers, and the evaluation has been overseen Satyajit Rath as the Senior Editor and Sara Sawyer as the Reviewing Editor.

The reviewers were very favorable about the manuscript. There are some small issues that need to be addressed before acceptance, as outlined below:

1) While the writing is quite good, the figures are poor. The legibility, labeling, and overall appearance of all figures needs to be improved before this work can be published anywhere, but in particular at a high-profile journal like *eLife*.

2) Could these antibodies be used as passive therapy in acute dengue cases? This should be addressed, because such an approach is extremely timely and is currently revolutionizing Ebola survival (look up drugs REGN-EB3 and mAb114 if not familiar, which are both monoclonal antibodies). One might even consider mentioning this very timely fact in the Abstract and how you have generated similar antibodies that may work for severe cases of dengue in the same way as they are working for Ebola.

3) Please spell out exactly how these discovered antibodies would assist in vaccine design, in the Discussion or elsewhere. I assume that the idea here is that the identification of this new epitope allows one to design a peptide vaccine? But then again, the epitope is at the intersection of monomers (a quaternary epitope), so how exactly would that work? I think it is important to not be hand-wavey about the significance of these types of studies, but to give some real meat about what one would do with this information. It is my understanding that the isolation of neutralizing antibodies has never yet actually informed the design of a successful vaccine. Am I wrong?

4) A gentle reminder that acronyms are convenient for writers, but inconvenient for readers. Please remove as many acronyms as possible, in particular CF (clonal family), SHM (somatic hypermutation), RVP (reporter virus particles), etc.

5) Figure 3, legend and accompanying main text, both need to better explain how the ADE assay works.

6) Introduction, third paragraph: The authors are critical of the lead DENV vaccine candidates for a number of stated reasons. Have the authors considered that neutralizing antibodies are not the only mediators of protection against dengue disease and that vaccines may need to be judged on more than Nabs?

7) Results, first paragraph: Probably best to disclose at this early point which patients were primary and secondary DENV cases and what is the secondary serotype (especially for 013). The fact that 013 is a secondary infection is important, but I didn't see this information until the Discussion.

8) Results, first paragraph: RVPs for all four DENV serotypes are used, but RVPs for DEN3 and DEN 4 are not listed in the Materials and methods. Also, why not use RVPs containing the contemporary strains rather the "lab strains" that you disparage in the Introduction?

9) Results, "Binding profile of mAbs" paragraph: J8 appears out of thin air – there is no clue from Figure 1—figure supplement 1 that J8 is worth including. It is not until the net page that the inclusion of J8 is described.

10) Results, "ADE potential of mAbs" paragraph: Why is ADE measured only for DENV2? Do you have data to suggest that the peak enhancement titer is the same across serotypes?

11) Discussion, second paragraph: This is where we learn that samples are from an acute secondary infection. This raises the question about what you would see in an acute primary infection? Are you only assuming what is alluded to in the fourth paragraph of the Discussion – that primary infection elicits low affinity cross-reactive antibodies? What about vaccination? Do you need cross-reactive antibodies if you have potent Nab induced against all four serotypes?

12) Discussion, last paragraph. For patient 013, high potency J8 and J9 bNAbs were elicited after the second infection, sometime during the course of severe disease onset. It is likely that these bNAbs appeared too late to mitigate the severe disease, unless the plasmablasts were isolated before disease presentation, which would be unusual. If this is the case, then the patient history and sampling times need to be very clearly described. I am assuming the normal course of events would be: primary infection (low affinity nAb, likely from an asymptomatic infection), second infection with severe disease (recall of nAb and increased affinity all identified in samples taken during clinical presentation), likely lifelong cross-protection (at the cost of severe disease).

13) Discussion, last paragraph. As supported by your finding, bNAbs may be a minor component of the overall polyclonal response. Please discuss the implications of this finding. Is this problematic if the polyclonal response also includes type-specific Nabs for DENV1-4? How does this influence the aims of vaccination? Should vaccines be designed to induce cross-protective Nab antibodies? Designed to induce homotypic Nabs to all serotypes? Both (probably the obvious answer)? Designed to induce other mediators of protection such as CD8 T-cells?

14) Materials and methods, RVP paragraph. Where is the description for DENV3 and DENV4 RVPs? Also, what are strain dates for the DENV isolates (how "uncontemporary" are they)?

15) Materials and methods, Contemporary DENV1-4 isolates paragraph. Are 2006 isolates still contemporary? Certainly they are "more contemporary" than any of the RVP strains, but wouldn't contemporary normally imply they are from the last decade? What are the dates for your contemporary DENV 1 and DENV4 strains?

16) Is the DENV serotype responsible for the primary infection of patient 013 known, or could it be inferred from analysis of PRNT responses of the acute phase secondary infection sera collected shortly after onset of symptoms versus later (convalescent) timepoint samples? Do the authors believe that any combination of serotypes causing heterologous primary and secondary dengue infections could give rise to similar broadly neutralizing antibody types, or could the specific combination of infecting serotypes be important?

17) It may be appropriate to also mention in the final paragraph of the Discussion that tertiary dengue infections are reported relatively frequently from studies looking at incidence and characteristics of sequential infections.

18) Please add the numbers of cells and quantities of RVPs (not just the calculated% final infectivity) used per experiment to the descriptions of neutralization and ADE assays in the Materials and methods section.

---

## [Author Response]

The reviewers were very favorable about the manuscript! There are some small issues that need to be addressed before acceptance, as outlined below:1) While the writing is quite good, the figures are poor. The legibility, labeling, and overall appearance of all figures needs to be improved before this work can be published anywhere, but in particular at a high-profile journal like eLife.

Thank you for this feedback. While we used precisely the suggested software listed above to prepare the original figures, we greatly appreciate the specific recommendations provided by the Reviewing Editor and have made the following suggested changes, which we hope improves their presentation:

All figures:

Improved font legibility and size uniformity (Arial 9pt or 10pt for most figures, with no more than a 2 point size range within each figure);

Removed excess white space as much as possible;

Removed acronyms as much as possible.

Figure 1C and Figure 1D:

Matched color of symbol border and curves to symbol fill colors to help distinguish binding curves;

Grouped items in figure key to loosely match grouping of binding curves.

Figure 2A:

Matched color of symbol border and neutralization curves to symbol fill colors to help distinguish curves;

Created a common x- and y-axis title for all graphs to reduce clutter;

Figure 2—figure supplement 1:

Changed neutralization curves and bar graphs from grayscale to color.

Figure 2—figure supplement 2:

Matched color of symbol border and neutralization curves to symbol fill colors to help distinguish curves;

Added panel C to indicate characteristics of DENV strains used to generate reporter and fully infectious viruses.

Figure 2—figure supplement 3:

Changed neutralization curves and bar graphs from grayscale to color.

Figure 4:

Added a common x-axis label to indicate DENV2 E mutation;

Added labels above each crystal structure to indicate the specific antibodies impacted by the mutations at the highlighted residues;

Removed grids from figure keys to reduce clutter.

Figure 6:

Rearranged figure panels to improve legibility and reduce white space;

Color coded highlighted clones along branch tips of phylogenetic tree in panel A to match sequence position in corresponding alignment in panel B;

Condensed the heavy and light chain amino acid sequence alignment in panel B to include mostly complementary determining regions; we have included nucleotide sequences of the entire heavy chain region in Figure 6—source data 1;

Highlighted key amino acid mutations in color to distinguish them from other mutations (shown in grey) relative to germline.

2) Could these antibodies be used as passive therapy in acute dengue cases? This should be addressed, because such an approach is extremely timely and is currently revolutionizing Ebola survival (look up drugs REGN-EB3 and mAb114 if not familiar, which are both monoclonal antibodies). One might even consider mentioning this very timely fact in the Abstract and how you have generated similar antibodies that may work for severe cases of dengue in the same way as they are working for Ebola.

We thank the reviewer for this suggestion, and have now added a caveated statement on the therapeutic potential of these antibodies in the second paragraph of the Discussion:

“In addition to informing vaccine design, J9 and J8 could serve as a foundation for the

development of monoclonal antibody therapy, which has recently revolutionized Ebola treatment (NIAID, 2019). However, the lack of a robust in vivomodel of dengue disease continues to hamper our understanding of dengue pathogenesis, and thus the assessment of therapeutic interventions (Chan, Watanabe, Kavishna, Alonso, and Vasudevan, 2015).”

3) Please spell out exactly how these discovered antibodies would assist in vaccine design, in the Discussion or elsewhere. I assume that the idea here is that the identification of this new epitope allows one to design a peptide vaccine? But then again, the epitope is at the intersection of monomers (a quaternary epitope), so how exactly would that work? I think it is important to not be hand-wavey about the significance of these types of studies, but to give some real meat about what one would do with this information. It is my understanding that the isolation of neutralizing antibodies has never yet actually informed the design of a successful vaccine. Am I wrong?

While it has been possible to develop successful antiviral vaccines without first characterizing antigen-specific monoclonal antibodies and their epitopes, many viral pathogens have resisted empirically developed candidate vaccines. We view our discovery and characterization of DENV bNAbs and characterization of their epitope as a first step towards rational epitope-focused vaccine design. As mentioned in our Introduction (sixth paragraph), unlike for other important viral pathogens such as HIV or influenza, we still lack a fundamental understanding of which regions on the DENV envelope protein can be targeted by broadly neutralizing antibodies. In our opinion, the finding that the bNAbs we discovered target a quaternary epitope is itself informative. As the reviewer pointed out, a peptide likely will not elicit these bNAbs. Rather, it would be necessary to design an immunogen that faithfully displays the epitope in its native quaternary conformation.

We have added the following sentences to the Discussion (second and sixth paragraphs), which we hope more clearly demonstrates how our findings can form the basis for future vaccine design efforts:

“Future studies relevant to vaccine development include detailed structural characterization to precisely reveal the epitope targeted by J9 and J8, and potentially protein engineering efforts for optimal epitope display (Graham et al., 2019). While the detailed characterization of neutralizing antibodies and their epitopes have not yet informed the design of a successful vaccine, there is promising proof of concept demonstrated for respiratory syncytial virus (Crank et al., 2019)”...

“Although based on the analysis of a single donor, one implication of our finding is that immunogen design to elicit bNAbs may require masking of immunodominant epitopes targeted by weakly neutralizing antibodies to focus the antibody response on the desired epitope (Weidenbacher and Kim, 2019).”

4) A gentle reminder that acronyms are convenient for writers, but inconvenient for readers. Please remove as many acronyms as possible, in particular CF (clonal family), SHM (somatic hypermutation), RVP (reporter virus particles), etc.

We agree that the overuse of acronyms is cumbersome and have removed as many acronyms in the main text as possible. In addition to the above, we have removed:

mAbs (monoclonal antibodies)

BCR (B cell receptor)

YFV (yellow fever virus)

JEV (Japanese encephalitis virus) rE (soluble recombinant E protein)

NGS (next-generation sequencing)

5) Figure 3, legend and accompanying main text, both need to better explain how the ADE assay works.

In addition to adding more details in the Materials and methods, we have now included a more detailed description of the ADE assay in the Figure 3 legend and main text. From the figure legend:

“Serial dilutions of the antibodies indicated above each graph were pre-incubated with (A) DENV2, (B) ZIKV or (C) WNV reporter virus for 1 h at room temperature prior to infection of K562 cells, which express FcgR and are poorly permissive for direct infection in the absence of antibodies. The y-axis shows the percentage of infected GFP-positive cells quantified by flow cytometry.”

6) Introduction, third paragraph: The authors are critical of the lead DENV vaccine candidates for a number of stated reasons. Have the authors considered that neutralizing antibodies are not the only mediators of protection against dengue disease and that vaccines may need to be judged on more than Nabs?

We chose to emphasize neutralizing antibodies in the Introduction both to highlight the fact that they are the focus of most DENV vaccines in advanced clinical development and to provide relevant context for the goal of our study, which was to identify and characterize broadly neutralizing antibodies. Nevertheless, we agree that protective immunity against DENV will most certainly involve additional immunological responses and have modified our ending to the Discussion as follows:

“The complex interplay among B cells and antibodies of different specificities and functions, and their impact on immunity and pathogenesis warrant further study, in addition to exploring other potential immunological correlates of protection (Katzelnick, Harris, and Participants in the Summit on Dengue Immune Correlates of, 2017), including cell-mediated (Ng et al., 2019; Rothman, Currier, Friberg, and Mathew, 2015) and non-neutralizing antibody responses (Sharma et al., 2019).”

7) Results, first paragraph: Probably best to disclose at this early point which patients were primary and secondary DENV cases and what is the secondary serotype (especially for 013). The fact that 013 is a secondary infection is important, but I didn't see this information until the Discussion.

We agree with this point and have now listed the DENV exposure history of the two patients in our current study in Table 1 and referred to it in the second sentence of the Results.

8) Results, first paragraph: RVPs for all four DENV serotypes are used, but RVPs for DEN3 and DEN 4 are not listed in the Materials and methods. Also, why not use RVPs containing the contemporary strains rather the "lab strains" that you disparage in the Introduction?

We have now added more detail for DENV3 and DENV4 RVPs in the Materials and methods and have also summarized the isolation dates of the reporter and fully infectious DENV strains used in our study in Figure 2—figure supplement 2C.

We did not attempt to generate RVPs encoding the structural genes of more contemporary DENV strains as not all combinations of flavivirus structural and non-structural gene components are compatible with the production of infectious RVPs (PMCID: PMC4524226). However, given that the use of RVPs simplifies the study of viral entry and its inhibition by antibodies, we agree that future efforts to generate RVPs encoding the structural genes of circulating strains are worthwhile.

9) Results, "Binding profile of mAbs" paragraph: J8 appears out of thin air – there is no clue from Figure 1—figure supplement 1 that J8 is worth including. It is not until the net page that the inclusion of J8 is described.

Thank you for pointing this out. As we mention both the binding and neutralization profile of J8 after its introduction in the section following the ‘Binding’ paragraph, we have now removed it from the latter.

10) Results, "ADE potential of mAbs" paragraph: Why is ADE measured only for DENV2? Do you have data to suggest that the peak enhancement titer is the same across serotypes?

As both antibody-mediated neutralization and enhancement are governed by a stoichiometric threshold of antibody binding (Pierson et al., 2007), we expected that antibodies with high neutralization potency will display low peak enhancement titers, as illustrated by our data for DENV2 in Figure 3. We have now extended the data supporting this hypothesis to include DENV1, DENV3, and DENV4 in Figure 3—figure supplement 1. Among all antibodies we tested, the peak enhancement titer is lowest for J9 and J8, consistent with their high neutralization potency.

11) Discussion, second paragraph: This is where we learn that samples are from an acute secondary infection. This raises the question about what you would see in an acute primary infection? Are you only assuming what is alluded to in the fourth paragraph of the Discussion paragraph – that primary infection elicits low affinity cross-reactive antibodies? What about vaccination? Do you need cross-reactive antibodies if you have potent Nab induced against all four serotypes?

As mentioned above, we have now described the DENV exposure history of the donor from which J9 and J8 were isolated in Table 1 and in the first paragraph of the Results. The statement that primary infection elicits mostly low affinity, weakly neutralizing cross-reactive antibodies is supported by multiple studies (reviewed in PMCID: PMC5116878 and cited in Discussion, fourth paragraph). As mentioned in the same paragraph, the few DENV broadly neutralizing monoclonal antibodies that have been described so far were isolated from individuals sequentially exposed to at least two DENV serotypes and it remains to be determined whether primary infection can elicit bNAbs. As for vaccination, broad and potent neutralization of all four serotypes could theoretically be mediated by either a polyclonal or monoclonal antibody response. However, as mentioned in the Introduction, the latter could potentially simplify vaccine design as it would be based on a single conserved immunogen (vs. four serotype-specific ones) to generate a balanced neutralizing antibody response to all four DENV serotypes.

12) Discussion, last paragraph. For patient 013, high potency J8 and J9 bNAbs were elicited after the second infection, sometime during the course of severe disease onset. It is likely that these bNAbs appeared too late to mitigate the severe disease, unless the plasmablasts were isolated before disease presentation, which would be unusual. If this is the case, then the patient history and sampling times need to be very clearly described. I am assuming the normal course of events would be: primary infection (low affinity nAb, likely from an asymptomatic infection), second infection with severe disease (recall of nAb and increased affinity all identified in samples taken during clinical presentation), likely lifelong cross-protection (at the cost of severe disease).

This is a good point, which we have now noted in the last paragraph of the Discussion. As we have indicated in Table 1, the first PBMC sample from which patient 013 plasmablasts were derived was isolated 4 days after fever onset. This patient presented with a diagnosis of dengue with warning signs and subsequently developed severe disease the next day.

13) Discussion, last paragraph. As supported by your finding, bNAbs may be a minor component of the overall polyclonal response. Please discuss the implications of this finding. Is this problematic if the polyclonal response also includes type-specific Nabs for DENV1-4? How does this influence the aims of vaccination? Should vaccines be designed to induce cross-protective Nab antibodies? Designed to induce homotypic Nabs to all serotypes? Both (probably the obvious answer)? Designed to induce other mediators of protection such as CD8 T-cells?

As discussed in point 6, we agree that protective immunity against DENV will likely involve multiple arms of the immune response. With regards to the direct implication of our finding above, we have added the following sentence to the last paragraph of the Discussion:

“Although based on the analysis of one donor, an implication of our finding is that immunogen design to elicit bNAbs may require masking of immunodominant epitopes targeted by weakly neutralizing antibodies to focus the antibody response on the desired epitope (Weidenbacher and Kim, 2019).”

14) Methods, RVP paragraph. Where is the description for DENV3 and DENV4 RVPs? Also, what are strain dates for the DENV isolates (how "uncontemporary" are they)?

We have now added more details for the DENV3 and DENV4 RVPs in the Materials and methods and also listed the isolation dates of the DENV strains in our study in Figure 2—figure supplement 2.

15) Methods, Contemporary DENV1-4 isolates paragraph. Are 2006 isolates still contemporary? Certainly they are "more contemporary" than any of the RVP strains, but wouldn't contemporary normally imply they are from the last decade? What are the dates for your contemporary DENV 1 and DENV4 strains?

Thank you to the reviewer for pointing this out. We agree that ‘more contemporary’ is a better description of the fully infectious relative to the RVP strains used and we have revised the text accordingly. As mentioned, in the Materials and methods and Figure 2—figure supplement 2, we have now also listed the year from which each of the DENV RVP and fully infectious strains was isolated (between 1964-1982 and 2004-2007, respectively).

16) Is the DENV serotype responsible for the primary infection of patient 013 known, or could it be inferred from analysis of PRNT responses of the acute phase secondary infection sera collected shortly after onset of symptoms versus later (convalescent) timepoint samples? Do the authors believe that any combination of serotypes causing heterologous primary and secondary dengue infections could give rise to similar broadly neutralizing antibody types, or could the specific combination of infecting serotypes be important?

This is an interesting question with important potential implications for vaccination strategies. The DENV serotype responsible for the primary infection is unknown and unfortunately, we cannot infer this information from the suggested experiment as the latest sample available for this patient is 22 days after onset of fever (Figure 5—figure supplement 4), which does not represent a truly convalescent timepoint. While specific combinations of infection by heterologous serotypes has been shown to be associated with severe disease (Alvarez et al., 2006 and Bhoomiboonchoo et al., 2015), whether the same is true for the development of bNAbs remains to be determined.

17) It may be appropriate to also mention in the final paragraph of the Discussion that tertiary dengue infections are reported relatively frequently from studies looking at incidence and characteristics of sequential infections.

We thank the reviewer for this suggestion, but think that this information might be more appropriate in the preceding paragraph in the Discussion where we referred to the potential role of sequential infections in the generation of bNAbs:

“To our knowledge, all human bNAbs against flaviviruses identified in the context of natural infection so far, including J9 and J8, were isolated from plasmablasts of individuals sequentially exposed to at least two DENV serotypes (Dejnirattisai et al., 2015; Xu et al., 2012; Zanini et al., 2018). We also cannot definitively rule out tertiary DENV infection, which is difficult to ascertain in endemic regions, given extensive cross-reactive antibody responses (Alvarez et al., 2006; Bhoomiboonchoo et al., 2015).”

18) Please add the numbers of cells and quantities of RVPs (not just the calculated% final infectivity) used per experiment to the descriptions of neutralization and ADE assays in the Materials and methods section.

We have now added this information in the Materials and methods.